# THE DECRYPTO BENCHMARK FOR MULTI-AGENT REASONING AND THEORY OF MIND

## ABSTRACT

We propose Decrypto, a novel interactive benchmark for evaluating coordination, competition, and theory of mind (ToM) reasoning capabilities in agentic, foundational AI models. Existing benchmarks often suffer from data leakage, saturation, and lack of interactivity, making it hard to measure the ability of intelligent systems to model other agents' reasoning. To overcome or alleviate these limitations, we introduce Decrypto, a multi-agent benchmark based on a popular, language-based board game and designed to be future-proof for large language models (LLMs). We validate Decrypto's effectiveness through comprehensive empirical evaluations of frontier LLMs, robustness studies, and human-AI cross-play experiments. We show that LLMs do not coordinate well with other LLMs or humans and perform strictly worse than the latter. Specifically, LLMs struggle to reason about the choices of others, even if they use the same underlying model, pointing to a fundamental limitation of current systems.

## 1 INTRODUCTION

Much recent effort has been made towards *agentic* behaviour and reasoning (Huang et al., 2023) to improve the capabilities of frontier foundational models. Multiple benchmarks have also been proposed to assess progress, with a focus on mathematical reasoning (Cobbe et al., 2021), common sense (Zellers et al., 2019), and theory of mind (Chen et al., 2024b).

However, many of those benchmarks suffer from significant shortcomings. For example, some benchmarks are based on a fixed dataset of problems. This leads to a significant risk of data leakage, whereas models will appear to perform well but become brittle when the questions are rephrased. Even when leakage risks are mitigated, such as having a secret test set, those benchmarks are subject to saturation. Other benchmarks for tasks such as maths (Cobbe et al., 2021), spatial reasoning Clark et al. (2018), or even multiple choice question answering (Hendrycks et al., 2020), target the weaknesses inherent in transformers trained on next-token prediction, such as failing to tokenize numbers correctly, inability to perform complex operations, and more. More importantly, real-world agentic applications are often multi-turn, multi-agent, partially observable, and stochastic. However, many of the supervised benchmarks do not address these specific challenges, unlike Decrypto.

Historically, games have proven valuable as benchmarks, requiring planning, decision-making, credit assignment and different types of reasoning. One such type of reasoning is theory of mind (ToM), which requires creating and maintaining a mental model of other agents (artificial or biological) within a multi-agent scenario. However, many ToM benchmarks are not interactive and suffer from one or more of the scenarios above. As a ToM task, Decrypto presents a multi-turn language variant of the Three Mountain Problem (Piaget et al., 1956) introduced to study child cognitive development. This seminal problem examines whether a child can acknowledge physical points of view distinct from its own. Additionally, Decrypto contains a cooperative and competitive aspect and thus requires players to reason about the information available to each of the other players, coordinate, and take strategic decisions accordingly.

Decrypto is inspired by the interactive environments prevalent in (multi-agent) reinforcement learning (RL) and based on the popular board game of the same name. In the paper, we first outline the design decisions to ensure that the benchmark is simultaneously future-proof and as *easy* as possible for LLMs, by removing every blocker that typically hurts LLM performance. We then introduce the different evaluation settings afforded by the game, evaluate a suite of popular frontier LLMs,

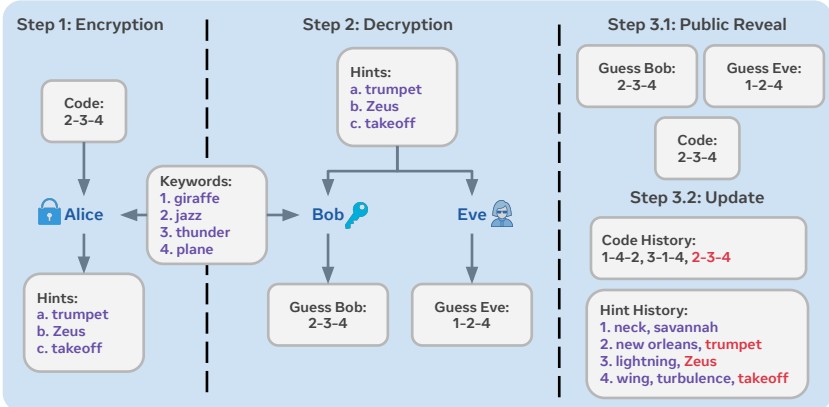

Figure 1: Overview of a turn in Decrypto, split into three steps, with Alice and Bob (Encoder and Decoder) playing against Eve (Interceptor). Step 1: Alice gets a random code of 3 non-repeating digits, and provides 3 hints referring to the meaning of the four *Keywords*. Step 2: Bob and Eve receive the hints and attempt to guess the code independently. Step 3: Both guesses and the code are publicly revealed, and the code and hint histories are updated accordingly. All players have access to the histories at all times, but only Alice and Bob have access to the keywords. The goal is for Alice to provide hints that are easy for Bob to decode but hard for Eve. As the game progresses, the growing hint history makes it easier for Eve to intercept.

perform a robustness study on our environment design, investigate human-AI cross-play results, and outline several exciting research directions enabled by our environment.

In summary, Decrypto distinguishes itself from existing benchmarks in several key ways:

1. **Focus on Language-Based Reasoning**: Unlike grid-world or embodied task environments, Decrypto isolates language-based reasoning and association, directly leveraging LLMs' core training objective. Decrypto is operated entirely through word associations, something that LLMs could reasonably be expected to excel at, since learning word co-occurrences is an important step towards reducing perplexity during pre-training.

2. **Theory of Mind Assessment**: Decrypto's design specifically targets theory of mind capabilities, requiring agents to reason about the knowledge of other players.

3. **Human-Compatible Evaluation**: As a game designed for humans, Decrypto facilitates comparisons between human and LLM performance, offering insights into the gap and compatibility between artificial and human intelligence in strategic communication tasks.

## 2 DECRYPTO: THE GAME

Decrypto is a word-based code-guessing board game published by Scorpion Masqué. It requires at least three players divided into two teams. We describe this variant here and implement it in this benchmark. Borrowing the terminology from cryptography, players are assigned three distinct roles: Alice (the Encoder), Bob (the Decoder), and Eve (the Interceptor or Eavesdropper). An instance of a game lasts 8 turns. We refer to one game instance as an episode, consistent with RL nomenclature.

As shown in Figure 1, Alice and Bob are on the same team and they share 4 ordered secret *keywords*, $\mathbf{k}^e = \{k_c\}^e$ where $c \in [1,4]$, $e \in \mathcal{N}$ is the current episode, and $k \in K$, where $K$ is a predetermined corpus of keywords. In the original game, the corpus contains around 450 keywords. For example, $\mathbf{k}^0$ might be $\{1.\ \text{guitar}, 2.\ \text{space}, 3.\ \text{apple}, 4.\ \text{sword}\}$. The keywords $\mathbf{k}^e$ are sampled at the beginning of the episode and remain fixed throughout the 8 turns.

Each turn, Alice samples a secret *code* of 3 non-repeating digits between 1 and 4, $\mathbf{c}^t = \{c_j\}^t$, where, $c \in [1,4]$ as above, $j \in [0,2]$, and $t \in [0,7]$, e.g. at turn 0, $\mathbf{c}^0 = \{4,1,3\}^0$. Alice must provide 3 public *hints*, one for each digit, $\mathbf{h}^t = \{h_c\}^t \ \forall c \in \mathbf{c}^t$, e.g. {knight, music, laptop} such that Bob can

guess the code but Eve cannot. Then, Bob and Eve make an independent attempt to **guess** the code, i.e., $\mathbf{g}_B^t = \{g_j\}_B^t$ and $\mathbf{g}_E^t = \{g_j\}_E^t$, where $g \in [1, 4]$, and the real code is revealed.

If Bob guesses incorrectly, his team gets a Miscommunication token. If Eve guesses correctly, she gets an Interception token. If, at any point in the episode, Alice and Bob accumulate two miscommunication tokens or Eve gains two Interception tokens, the episode ends, and Eve wins. Alice and Bob win if they make it through 8 rounds without any of those two conditions happening.

Bob starts with an advantage because Bob has access to the 4 keywords $\mathbf{k}^e$ and Eve does not, see Figure 1. However, the actual code is revealed publicly after Bob and Eve provide their guess. Eve can, therefore, keep track of the hint history $\boldsymbol{\tau}^e = \{\tau_c\}^e$, where $c \in [1, 4]$ and $\tau_c^t = \{h_c^0, ..., h_c^t\}$, i.e., the hints used for each digit on previous turns. $\boldsymbol{\tau}$ makes it easier to intercept the code as the game progresses. For instance, if Alice provided the hints $\tau_4^3 = \{$knight, duel, Middle Ages, blacksmith$\}$ for digit 4 in the previous 4 turns, and now gives the hints $\mathbf{h}^{t=4} = \{$"shield", "accountant", "snow"$\}$, it is quite likely that Eve will associate "shield" to digit 4. Therefore, Alice must be careful to provide hints that are subtle enough to avoid interception yet sufficiently related to the keywords for Bob to guess correctly.

While the keywords are sampled from a predetermined set, *the choice of hints is open-ended*. The restriction is that hints must be real words (including proper nouns) and refer to the meaning of the keywords, not to their spelling or pronunciation.

The game provides a language reasoning challenge. Alice must choose associated hints carefully, using theory of mind to anticipate how Bob and Eve will interpret them using their respective available information. Both miscommunications and interceptions are detrimental.

# 3  DECRYPTO: THE BENCHMARK

Unlike other reasoning benchmarks, Decrypto is purposely designed to *not* require many of the capabilities that large pre-trained models struggle with. In particular, it demands no symbolic reasoning (Clark et al., 2018; Bard et al., 2020), mathematical reasoning ((Cobbe et al., 2021; Zhang et al., 2024a), spatial reasoning (Clark et al., 2018; Carroll et al., 2019), tool use (Xu et al., 2023a), or particular attention to tokenization (which has notably been shown to affect arithmetic performance).

Instead, strong performance in Decrypto relies purely on word-based reasoning and theory of mind. Because LLMs are trained to learn word co-occurrences and associations, we expect them to excel at the game. However, this is not the case, and humans and simple hard-coded baselines outperform even the most advanced open—and closed-source LLMs available.

Due to its two-team, three-player setup, Decrypto can be used to benchmark LLMs in both competitive and cooperative scenarios. Being a game, the difficulty of Decrypto naturally scales with the agents' ability for each of the three roles. This makes the benchmark much harder to saturate, unlike those relying on a fixed dataset of problems. We also curate a set of 680 possible keywords, resulting in over 8.8 billion possible keyword combinations to limit memorization, and implement the option to provide custom keywords beyond the standard set. Furthermore, the game was designed for human play, facilitating human-AI coordination and qualitative performance comparison.

The benchmark, including integration with popular APIs, all keywords, and helper code to run experiments and collect human data will be open-sourced along with this paper's final release.

**Competition.**  The first aspect of the Decrypto benchmark is to evaluate language reasoning in a competitive setting. This setting involves assigning Alice and Bob with identical agents (i.e. the same LLM) and evaluating the Alice-Bob pair against Eves with different agents. The goal is to determine how well an LLM can play with itself to win against various interceptor Eves. A model that outperforms other LLMs as Alice-Bob, and as Eve, likely has stronger reasoning capabilities. The average number of turns per game is the most predictive performance variable.

The key metrics provided naturally by the game are the number of interceptions, the number of miscommunications and certainly the number of wins either by Alice/Bob or Eve. The number of miscommunications measures the ability of Alice and Bob to cooperate and the number of interceptions captures Eve's ability to compete. There is a tension between miscommunications and interceptions in Decrypto. Alice can aim to minimise miscommunications and provide easy hints.

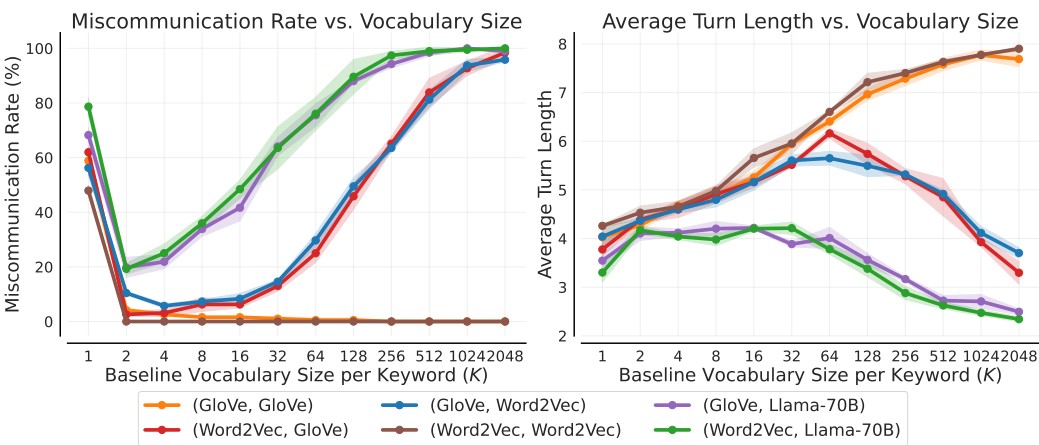

Figure 2: Percent of games ending in miscommunications (left) and average game length (right) for the word embedding baselines as a function of $K$ – the hint vocabulary size per keyword. For low $K$, the baselines coordinate well, but their hints are easier to intercept. At high $K$ values, both models sample from lower similarity words. This results in very strong SP baselines, which cannot coordinate in XP, with most games ending in miscommunication for $K > 128$. Each curve corresponds to an (encoder, decoder) pair. The interceptor is Llama3.1 70B, but trends hold across all interceptors, including baselines. We report mean and standard error over 3 model seeds.

If Alice's hints were easy to guess, Alice would never miscommunicate with Bob, but Eve could certainly intercept the code. If Alice aims to minimise intercepts, the hints would be impossible to guess for Eve, but also for Bob. Thus, the number of miscommunications and intercepts are two sides of the same coin. Average game length per game allows us to capture both sides in one metric. If the average length is high, Alice and Bob can successfully balance the difficulty of hints to avoid miscommunications and intercepts. The number of interceptions and miscommunications only capture part of the game. On one hand, if Alice wanted to avoid miscommunications, she would provide obvious hints but get intercepted quickly. On the other hand, Alice could give hints that are almost impossible to guess to avoid interceptions. In both cases, the games would be over quickly soon, so the number of turns is an insightful metric.

Benchmarking LLMs directly with and against other state-of-the-art LLMs is akin to Chess or Go, where the best models are evaluated against each other and not on a fixed dataset of trajectories. We strongly believe that solving the Decrypto challenge likely requires LLMs to perform multi-step reasoning and to train them in self-play (SP) for goal-oriented tasks.

**Ad-hoc Coordination** In this setting, we are interested in evaluating the coordination ability. This setting freezes Eve (e.g., to a rule-based baseline or the strongest available LLM). It then pairs different LLMs with each other (e.g. Alice is GPT-4o, and Bob is Llama). What matters is the ability to coordinate with previously unseen agents. As above, the average number of turns per game is the most reasonable metric here.

A subset of ad-hoc coordination is **human-AI coordination**. This setting is similar to the above, except one of the two agents (Alice or Bob) is played by a human. In this case, we evaluate the ability of LLMs to coordinate with humans, understand why they gave certain hints, and anticipate how they might interpret given hints.

**Theory of Mind** Both the competitive and cooperative settings require theory of mind for high performance. Each of the three players has access to different information. Bob and Eve must approximate or recreate the reasoning that Alice took to arrive at her proposed hints to guess correctly. Meanwhile, Alice must anticipate how each of the other two players will interpret the hints and choose accordingly.

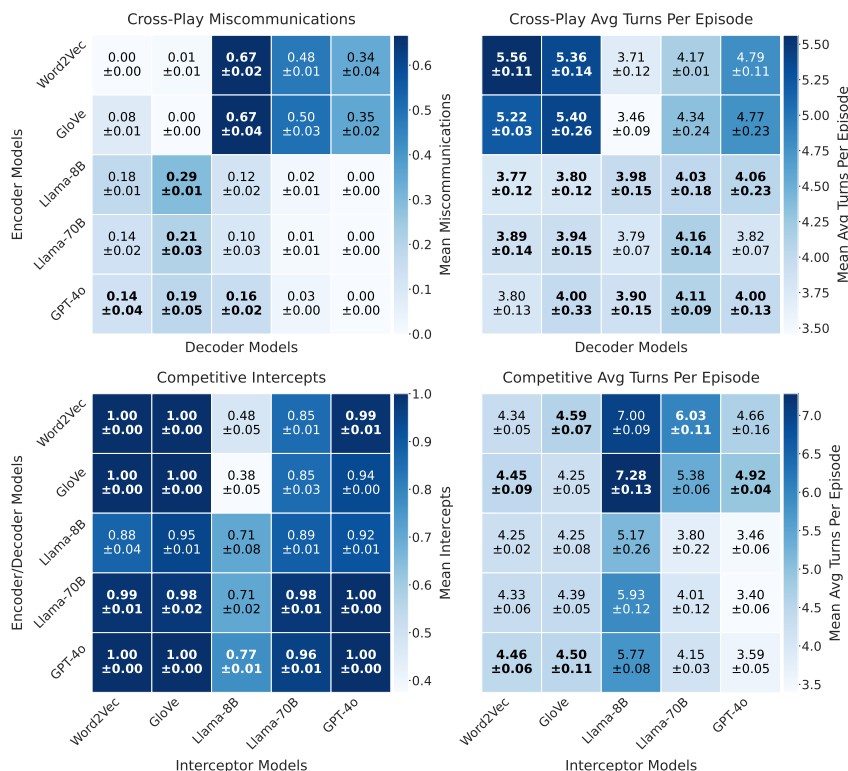

Figure 3: Cooperative and competitive results for 5 different agents. All results were reported for 32 games and 3 model seeds. **Top:** Cooperative cross-play matrix when the interceptor is Llama3.1 70B. *Left:* proportion of games ending in miscommunication. *Right:* average number of turns per episode. At K=16, baselines play well with each other, outperforming all LLM agents tested, but coordinate poorly with LLMs. **Bottom** Competitive results playing a homogeneous encoder-decoder team against an interceptor. *Left:* proportion of games ending in an intercept. *Right:* average number of turns per episode. Here, too, baselines outperform LLM agents across both roles.

# 4 STATE OF THE ART

## 4.1 SPECIALIST VS GENERALIST AGENTS

To avoid trivial solutions for our benchmark, we will discuss two different types of agents, **specialist agents** and **generalist agents**, and highlight which directions we encourage for exploration. In most RL settings, the policy is designed specifically for a given environment since it is parameterised to map the environment's observations (or action-observations history) to its action space. This limitation no longer necessarily holds with general-purpose models, mainly because large pre-trained models are expected to perform well in several scenarios beyond their training distribution (Brown, 2020). As a result, we propose to divide agents into two distinct classes: **specialist agents** and **generalist agents**.

**Specialist agents** are all agents purposely designed using knowledge of the task. Solutions in this space include rule-based strategies, fine-tuning a large pre-trained model on game data, building a prompt pipeline informed by the rules of the games, and prompt engineering. In this framework, we *consider the prompt to be part of the agent*, if it is needed at all. In this context, prompt engineering is the designer's burden, and the environment only returns key information, such as a dictionary.

Due to the larger design space, we expect specialist agents to outperform generalist ones in every task. However, we especially encourage research on methods which produce specialist agents for a large class of tasks. This includes better fine-tuning algorithms but excludes rule-based agents for playing Decrypto, such as the Word2Vec (Mikolov, 2013) and GloVe (Pennington et al., 2014) baselines we present next.

**Specialist Agents - Baselines** We introduce two specialist agent baselines to play Decrypto. For the baseline models, we use the common crawl GloVe with a vocabulary of 2.2 million words (Pennington et al., 2014) and the Word2Vec embeddings trained on the Google News dataset, provided by the gensim library (Rehurek & Sojka, 2011; Mikolov, 2013). The baselines work as follows. Alice can choose hints from a fixed corpus of 5696 hints. The hint corpus comprises the most common nouns extracted from the Brown, Gutenberg, and Webtext corpora (Bird et al., 2009). We filter the nouns to ensure all extracted nouns are present in the GloVe and Word2Vec embeddings. (We use this hint corpus for the baselines only; LLM agents generate hints in an open-ended fashion, limited only by the game rules and the model itself.)

Before Alice chooses hints, Alice and Bob agree on a strategy. For each code digit $c_j^t$, Alice picks a hint $h_c^t$ and guarantees that the hint picked for the digit is more similar to the code's keyword $k_c$ than to any other keyword, i.e., $s(h_c^t, k_c) > s(h_c^t, k_j)$ $\forall j \in -\mathbf{c}$, where $-\mathbf{c} = [1, 4] \setminus c$ and $s$ is the cosine similarity.

Alice increases the diversity of her hints by randomly sampling each hint from the top-$K$ most similar words for each keyword under cosine similarity, filtering words that do not satisfy the constraint above. We also enforce that Alice cannot reuse any previously used hints from the same episode.

Bob then analyses the hints and assigns each hint to its most similar keyword under cosine similarity, i.e., $\arg\max_{c \in C} s(h_X, K_c)$ $\forall X \in \mathbf{X}$. Bob guesses the code perfectly if Alice and Bob use the same word embeddings because the hints are guaranteed to be the most similar to the correct keyword under the same embedding model.

Eve is only given the hints $\mathbf{h}$ and the hint history for each keyword, which at turn 0 is empty. Eve calculates the cosine similarity between the average embedding of each keyword's hint history with the hints. This results in a similarity matrix of size $N \times M$, where $N(= 3)$ is the number of hints and $M(= 4)$ is the number of keywords. Eve combinatorially calculates the globally optimal guess based on the similarity values. Since $N$ and $M$ are typically small values, we are unconcerned about the computational complexity. However, Eve's selection task is equivalent to a linear assignment problem and efficient algorithms, such as the Jonker-Volgenant algorithm, are available in open-source libraries such as SciPy (Virtanen et al., 2020).

These baselines serve three purposes. First, they demonstrate that by pre-agreeing on a strategy and having perfect theory of mind, represented by a shared word similarity measure, it is possible to construct a virtually unbeatable Decrypto team, in the role of Alice and Bob, by setting $K$ to a sufficiently large value. This establishes an upper bound on performance to which to compare other agents. Second, while they achieve arbitrarily good performance, we show in Figure 2 that changing the similarity measure for one of the agents can result in a catastrophic lack of coordination and poor cross-play (XP) performance. Third, by selecting a low enough value of $K$, we can operate the baselines in the regime where the two embeddings (Word2Vec and GloVe) correlate and are likely to still rely on the useful signal captured by those embeddings. For this reason, we use baselines with $K = 16$ for the remainder of the paper, corresponding to the point in Figure 2 before miscommunications increase significantly.

**Generalist agents**, in contrast to specialised agents, refer to general-purpose models used to play the game "out-of-the-box" without any additional fine-tuning on task-specific data. In particular, for generalist agents, we *consider the prompt to be the observation* returned by the environment, which precludes any form of prompt engineering. Evaluating foundational models in this framework assesses their ability to generalise to novel tasks, including out-of-distribution ones and those where data is limited or unavailable.

Because foundational models are sensitive to their prompting, we recommend evaluating generalist agents with a range of different prompts, to prevent the choice of prompt favouring any one model. We provide such an evaluation using various hand-crafted Decrypto prompts in Figure 4 and show that game performance is robust to prompt variations and depends significantly on model size.

Generalist agents include individual pre-trained models and multi-agent systems of LLMs or tool-augmented generation. The only requirement is that any such agent be defined at a general level of abstraction and not condition in any way on the evaluation benchmark (e.g., the Decrypto game). For instance, an LLM that automatically rephrases the prompt for clarity is acceptable. Using a regular expression to extract key information from the Decrypto prompt is not.

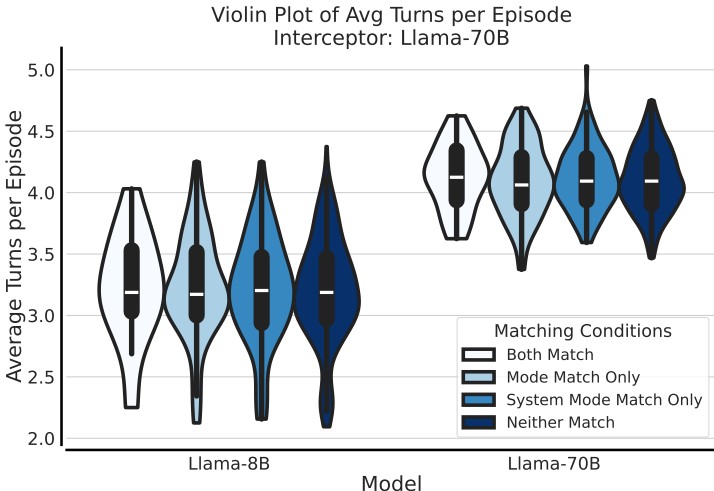

Figure 4: Distribution of the number of turns per game as we vary the system and user prompts for both encoder and decoder. We handwrite 5 system and 5 user prompts (i.e. "modes") for the encoder and the decoder and plot the distributions of games when only their system prompts match, when only their user prompts match, when both match or when neither do (625 total combinations per model). We find that both models tested are robust to significant prompt variations, with model size remaining the most significant predictor of performance.

## 4.2 HUMAN-DATA COLLECTION

We collect 9 full games of human ad-hoc cooperative trajectories against a fixed LLM Eve, namely Llama-3.1-70B. The human players interacted with the game through a unified commandline interface (see Appendix B.2 and saw the same system and user prompts, and information as an LLM would. Please see Section A.2,A.3 for example prompts.

Normally, a game ends when 2 interceptions or miscommunication tokens are collected. To maximise our data collection, we did not terminate the game after 2 interceptions but let the humans play the game for the full 8 turns. The participants were incentivised to keep providing good hints and analyse the hints appropriately even after the game technically terminated. Such a game does not count as won, but the extra turns can be useful when replaying the game to evaluate other LLMs as Eve.

## 5 RESULTS

To kickstart this benchmark, we assess the cooperative, competitive, and theory of mind capabilities of specialised and generalist agents in Decrypto. Amongst generalist, open-source models, we evaluate, in order of parameter count, Llama-3.1 8B, Llama-3.1 70B, Mistral-Large-Instruct-2407 (123B). From the closed-source models, we only evaluate GPT-4o due to resource constraints. Moreover, we include two specialist agent baselines based on word embedding models, which we describe below. Finally, we present the human-AI coordination and competition results, with data collected from 9 human games.

**Baselines.** We first look at the specialised agents baselines and show in Figure 2 that we can control $K$ for the top-$K$ selection to make them arbitrarily strong. Smaller $K$ prioritises words semantically similar to the keyword, making the hints easier to guess. Bigger $K$ might select semantically less similar hints, which are harder to guess if Bob cannot access the same embeddings. For example, GloVe-Alice wants to hint the keyword "fire" to Word2Vec-Bob, and $K = 50$. The 50th most similar word to "fire" in the hint corpus would be "oil" for GloVe. However, "oil" is not even in the top 1000 most similar words for Word2Vec-Bob in the hint corpus, highlighting where cross-play

| Role | Model | Events | Event Rate | Surv. Rate | Avg. Game Len. |
|---|---|---|---|---|---|
| Interceptor | Word2Vec | $7.00 \pm 0.00$ | $11.29\% \pm 0.00\%$ | $42.86\% \pm 0.00\%$ | $6.89 \pm 0.00$ |
| | GloVe | $7.00 \pm 0.00$ | $12.07\% \pm 0.00\%$ | $\mathbf{28.57\% \pm 0.00\%}$ | $6.44 \pm 0.00$ |
| | Llama3.1 8B | $4.00 \pm 0.58$ | $6.56\% \pm 1.07\%$ | $50.00\% \pm 4.12\%$ | $6.81 \pm 0.13$ |
| | Llama3.1 70B | $6.67 \pm 0.88$ | $10.91\% \pm 1.59\%$ | $45.24\% \pm 2.38\%$ | $6.81 \pm 0.13$ |
| | Mistral Large | $8.67 \pm 0.33$ | $14.32\% \pm 0.84\%$ | $38.43\% \pm 3.24\%$ | $6.74 \pm 0.16$ |
| | GPT-4o | $\mathbf{9.67 \pm 0.67}$ | $\mathbf{17.46\% \pm 1.68\%}$ | $41.67\% \pm 4.17\%$ | $\mathbf{6.19 \pm 0.20}$ |
| Decoder | Word2Vec | $18.00 \pm 0.00$ | $75.00\% \pm 0.00\%$ | $0.00\% \pm 0.00\%$ | $2.67 \pm 0.00$ |
| | GloVe | $18.00 \pm 0.00$ | $62.07\% \pm 0.00\%$ | $0.00\% \pm 0.00\%$ | $3.22 \pm 0.00$ |
| | Llama3.1 8B | $18.00 \pm 0.00$ | $70.15\% \pm 0.92\%$ | $0.00\% \pm 0.00\%$ | $2.85 \pm 0.04$ |
| | Llama3.1 70B | $15.00 \pm 0.00$ | $34.40\% \pm 0.93\%$ | $7.41\% \pm 3.70\%$ | $4.85 \pm 0.13$ |
| | Mistral Large | $16.00 \pm 0.00$ | $34.78\% \pm 0.00\%$ | $12.50\% \pm 0.00\%$ | $5.11 \pm 0.00$ |
| | GPT-4o | $16.67 \pm 0.33$ | $39.42\% \pm 1.08\%$ | $0.00\% \pm 0.00\%$ | $4.70 \pm 0.15$ |
| | Human | $\mathbf{10.00 \pm 0.00}$ | $\mathbf{16.39\% \pm 0.00\%}$ | $\mathbf{33.33\% \pm 0.00\%}$ | $\mathbf{6.78 \pm 0.00}$ |

Table 1: We collect 9 games from human encoder-decoder teams and report the agents' performance when playing as (top) interceptor against human players or as (bottom) decoder with a human encoder. Events are interceptions (higher is better) when the role is Interceptor and miscommunications (lower is better) when the role is Decoder. Of all the agents tested, we see that GPT-4o is the strongest interceptor against humans. In ad-hoc human team-play, all agents tested underperform compared with humans, getting significantly more miscommunications, leading to shorter games. All agent results report mean $\pm$ standard error over 3 seeds.

(XP) difficulties arise. This finding also holds when *LLMs* play against baselines. As $K$ increases, the miscommunications increase, and the average turn length thus decreases, as shown in Figure 2.

**Crossplay.** Baseline-LLM teams get significantly more miscommunications than baseline-baseline or LLM-LLM teams. The top row of Figure 3 shows the total number of games ending in miscommunication out of 32 games for the two baselines and two LLMs, Llama-3.1-8B and Llama-3.1-70B. Among LLMs, the main determining factor for miscommunications is the model used for Bob, with the smallest model, LLama-3.1-8B, seeing the most miscommunications. *Interestingly, we do not observe any Self-Play/Cross-Play gap*, even though it would technically be possible for an LLM to perfectly model its counterparts when playing with or against the same model. Moreover, note how very low miscommunications, e.g., between GPT-4o and Llama-3.1-70B, do not significantly improve game length, i.e., number of turns per game.

**Competitive.** For our competitive results, larger models generally perform better as both Alice/Bob and as Eve. However, we find that the win rate is heavily skewed in favour of Eve, as shown in Figure 5, with most models rarely surviving. We measure the number of interceptions and average game length of different SP teams against different Eve Agents. We report our results in the bottom row of Figure 3. However, as our human experiments show, such a heavy bias is not a property of the game but instead of the LLMs themselves. Indeed, we find that humans achieve 33% win rate against even the strongest Eve agents.

**Robustness.** Next, we show that prompt variants do not significantly affect the final performance measured by average turn length for Llama-3.1-8B and Llama-3.1-70B, see Figure 4. This suggests that the poor performance of LLMs in Decrypto is more likely due to a lack of reasoning abilities than a lack of prompt tuning. We look at the robustness of different models to variations in prompts and generation parameters, keeping in mind that generalist agents cannot control their prompt since it is assumed to be part of the environment. We handcraft 5 system and user prompts for Alice and Bob, respectively. The system prompt consists of 2 components. One component is responsible for explaining the game rules in general, for which we have 5 variants. The second component explains the specific role. We have 5 prompt variants for each role, already resulting in 125 different prompt combinations. The user prompt instructs the specific roles to take their actions, for which we have 5 variants. In total, this results in 625 different prompt setups for each model. We run 32 games over 3 model seeds per system/user prompt combination and measure the average game length.

**Human Evaluation.** We demonstrate that LLMs perform worse than humans in Decrypto and that human hints are on par with the specialised baseline agents in Table 1 when competing with an LLM

| Encoder Model | Interceptor: Llama-8B | | Interceptor: Llama-70B | |
|---|---|---|---|---|
| | Total Predict | Intercept Predict | Total Predict | Intercept Predict |
| Llama-8B | $0.17 \pm 0.01$ | $0.47 \pm 0.01$ | $0.26 \pm 0.01$ | $0.45 \pm 0.03$ |
| Llama-70B | $0.17 \pm 0.01$ | $0.38 \pm 0.02$ | $0.25 \pm 0.01$ | $0.35 \pm 0.02$ |

Table 2: Theory of Mind Evaluation: We ask Alice to predict what Eve will guess. We report *Total Predict*, the total prediction accuracy of Alice, averaged across all turns and *Intercept Predict*, the prediction accuracy only for turns on which Eve successfully intercepts. **Alice struggles to predict Eve's guess, even when the same LLM plays both.** Alice has complete knowledge of the information available to Eve, demonstrating the limited ability of LLMs to model and reason about other agents' points of view.

Eve. First, humans have the lowest miscommunication rate at 16%, with Mistral-Large coming in second at 34%, thus more than double. Humans also have the highest survival rate at 33.33%, and again, Mistral comes second at 12.5%, and most other models never win at all. Furthermore, when different LLMs are matched against the human collected data, we achieve an average game length between 6.33 and 6.94, which puts humans on par with baselines' self-play when paired against the weakest LLM-Eve. These results provide strong evidence that LLMs lack the reasoning abilities to understand human hints, even though human hints work well with other humans. Human data collection details are in appendix B.1

**Theory of Mind**  The relatively weak performance of LLMs in the settings above are evidence that LLMs do not possess the ToM reasoning abilities necessary to play Decrypto well. This is illustrated in the failure cases in Appendix C. In one example, Alice fails to sufficiently reason about the difficulty of their hints and is easily intercepted. In another, Alice provides an ambiguous hint and fails to predict what Bob might guess.

Additionally, Decrypto provides a platform on which to conduct explicit ToM experiments inspired by works in cognitive psychology. The first such experiment is a word-based code-guessing variant of the Three Mountain Problem (Piaget et al., 1956). We explicitly ask Alice to predict what Eve will guess based on Alice's hints. We evaluate the prediction abilities of Llama-3.1-8B and Llama-3.1-70B as Alice and Eve each. We find that the accuracy is very low for both Alice models and that the larger Eve model is more predictable, as shown in Table 2. We also find that predictability and intercept ratio are closely connected, suggesting that a more capable Eve becomes more predictable.

The second experiment evaluates *representational change* (RC) and *false belief* (FB) in the way defined in the seminal work of Gopnik & Astington. Our procedure is the following: At each turn except the first, we branch out the context of the agent and prompt it three times independently. The first prompt asks it to predict the four keywords. The second prompt reveals the keywords and asks the model what it thought were the keywords before the reveal. The third prompt again reveals the keywords and asks the model to predict what a "second Interceptor" who has seen everything except the reveal would think are the keywords. By comparing the first and the second answers, we measure RC, the ability of the agent to recognize when its belief about the world (but not the world itself) changes due to additional information. By comparing the first and third answers, we measure FB, the ability to represent other agents as having inaccurate beliefs about the world. For the *Strong* variant of those tasks, we consider the agent to pass if it correctly predicts what it answered in question 1. We consider an agent correct for the *Weak* variant if the answers to questions 2 or 3 are not the real keywords. Results in Table 3 show that ability correlates with model size but that neither of the models gets perfect scores. On Strong tasks, pass rates are particularly low, evidence that LLMs only do not have persistent models of their "mind" of that of others.

Overall, the results strongly support that LLMs struggle to model others' reasoning and that Decrypto has the potential to be a fruitful benchmark for further exploring these capabilities.

## 6 RELATED WORK

Recent research has seen a surge in developing game-based environments and benchmarks to evaluate LLMs in multi-agent scenarios. These works span various domains, from grid-based worlds to

social deduction games, each offering unique insights into LLM capabilities. Our Decrypto benchmark builds on and differentiates itself from these existing approaches.

In multi-agent game environments, several frameworks have emerged. BattleAgentBench (Wang et al., 2024) and AgentBench (Liu et al., 2023b) introduce grid-based worlds to assess cooperation and competition. While comprehensive, their reliance on spatial reasoning makes them less suitable for evaluating the language-based deception and coordination central to Decrypto. LLM-Arena (Chen et al., 2024a) and GameBench (Costarelli et al., 2024) offer more diverse game sets, including word-based games like Undercover. These share similarities with Decrypto but lack its focused evaluation of theory of mind and coordinated deception in a purely language-based setting.

Social deduction and negotiation games have been another avenue for exploring LLM capabilities. Studies on Werewolf (Xu et al., 2023b;c), AvalonBench (Light et al., 2023), and ToMBench (Chen et al., 2024b) evaluate strategic communication and theory of mind, while LLM-Deliberation (Abdelnabi et al., 2023) examines interactive multi-agent negotiation. Decrypto builds on these approaches by providing a more structured environment for isolating specific language-based reasoning and coordination aspects.

Research has also explored LLMs in both cooperative and competitive dynamics. Studies on human-AI coordination in Overcooked (Liu et al., 2023a) and consensus-seeking in multi-robot tasks (Li et al., 2023) focus on cooperation. In contrast, AgentPro (Zhang et al., 2024b) examines competitive scenarios in games like Blackjack and Poker. Decrypto bridges these approaches by simultaneously evaluating cooperative and competitive dynamics within a single, language-centric framework.

Recent work has begun exploring theory of mind capabilities in LLMs, with studies like Guo et al. (2024) investigating this in embodied tasks within grid worlds. Broader evaluation frameworks such as Smartplay (Wu et al., 2023) offer comprehensive benchmarks across multiple games and capabilities. While valuable, these approaches often make isolating specific aspects of language-based reasoning and coordination challenging.

## 7 CONCLUSION

This paper introduces Decrypto, a novel benchmark designed to evaluate coordination, competition, and theory of mind capabilities in language models. Decrypto addresses critical limitations of existing benchmarks, such as data leakage and lack of interactivity, providing a more robust framework for assessing AI systems' reasoning and behaviour.

Our experiments, encompassing a range of open-source and closed-source language models, and word embedding systems, revealed that even state-of-the-art models struggle with the nuanced communication and strategic reasoning Decrypto requires. This highlights a significant gap between human-level theory of mind capabilities and current AI systems.

Decrypto offers a more explicit assessment of language models' core communication skills by isolating language-based reasoning from other factors like spatial reasoning or mathematical ability. The benchmark's adaptable difficulty, achieved through varying opponent sophistication and keyword complexity, alleviates the saturation issues common in fixed-dataset benchmarks. Our human-AI cross-play experiments shed light on the potential and limitations of human-AI coordination in strategic communication tasks.

Decrypto opens up several promising research directions. These include enhancing language models' theory of mind capabilities, using the benchmark as a training environment for multi-agent reinforcement learning, and examining the ethical implications of improving AI's strategic communication abilities.

In conclusion, Decrypto significantly advances our ability to assess and improve AI's agentic capabilities. It provides a challenging yet focused benchmark for multi-agent communication and reasoning, paving the way for more socially intelligent AI to better understand and interact with humans in complex strategic scenarios. As AI capabilities continue to evolve, Decrypto will serve as a valuable tool for measuring progress and identifying areas for improvement.

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

# A APPENDIX

## A.1 ADDITIONAL RESULTS

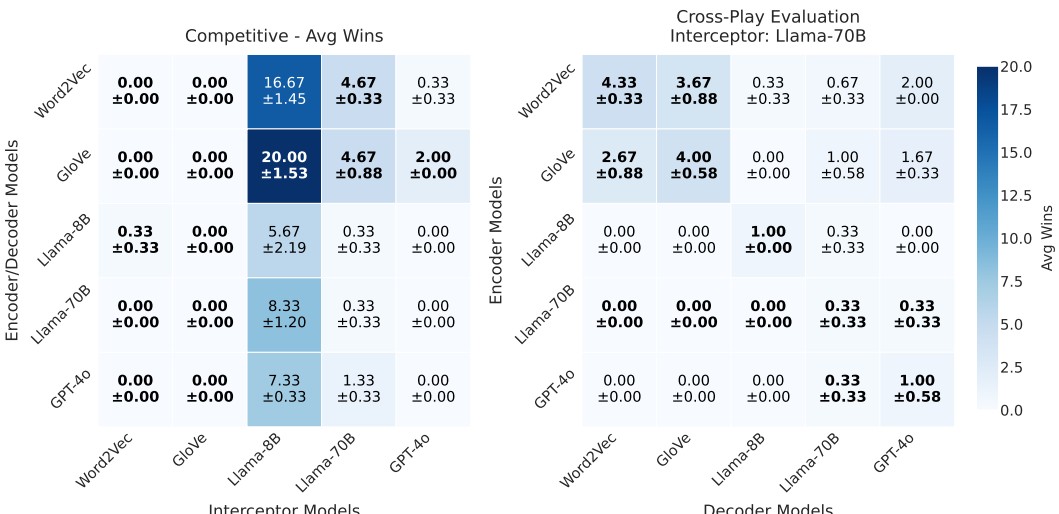

Figure 5: Number of survived games over 32 games over 3 model seeds.

| Interceptor Model | Strong | | Weak | |
|---|---|---|---|---|
| | Repr. Change | False Belief | Repr. Change | False Belief |
| Llama-8B | $0.05 \pm 0.02$ | $0.00 \pm 0.00$ | $0.70 \pm 0.01$ | $0.41 \pm 0.07$ |
| Llama-70B | $0.17 \pm 0.08$ | $0.00 \pm 0.00$ | $0.95 \pm 0.02$ | $0.89 \pm 0.03$ |

Table 3: ToM results evaluating *representational change* (RC) and *false belief* (FB) pass rates of two models. RC measures whether agents can accurately identify a change in their own view of the world following new information. FB measures whether agents can model the incorrect beliefs of another agent. Strong forms require to accurately infer the wrongly held beliefs. Weak forms only require agents to realize that their past belief or the belief of others does not match ground truth. Model size correlates with better *Weak* ToM, but both models perform poorly on the Strong task. Pass rates are computed over 64 episodes with varying lengths. We report mean and standard deviation over 3 seeds.

## A.2 SYSTEM PROMPTS

```
1   You are playing a variant of the code guessing ga e Decrypto. The setup
    of the game is the following:
2   The game is played with three players split into two teams.
3   The first team is composed of the Encoder and the Decoder.
4   The second team is composed of the Interceptor.
5   The Encoder and Decoder share a list of 4 secret keywords that they can
    rely on to help co munication. Each keyword has a digit associated with
    it.
6   For example, if the keywords are {airplane, guitar, hat, plant}, the
    digits associated with them would be {1, 2, 3, 4}.
7   The keywords are fixed for the entire game, so the Encoder and Decoder
    must not reveal them.
8
9   Each turn, the Encoder gets a secret code of 3 non-repeating digits
    between 1 and 4, e.g. 3-1-2.
10  They have to then provide 3 hints (one for each code digit) such that
    the Decoder can guess the code but the Interceptor cannot.
11  Then, the Decoder and the Interceptor make their guesses and the code is
    revealed.
12
13  The Interceptor wins an Interception token if they successfully guess
    the code.
14  The Encoder and Decoder get a Miscommunication token if the Decoder
    fails to guess the code.
15  The Interceptor wins if it gets two Interception tokens, or if the
    Encoder and Decoder get two Miscommunication tokens.
16  The Encoder and Decoder team wins if it makes it through 8 rounds
    without triggering two Miscommunications or two Interceptions.
17
18  The hints must be one or two words, and make a reference to the meaning
    of the secret keyword of the corresponding digit.
19  Codes are unique; the same code cannot occur twice in the same game.
20
21  For example, if the four secret keywords are
22  {1: "airplane", 2: "guitar", 3: "hat", 4: "plant"}
23  and the code for that turn is 3-1-2, which means that we need hints for
    {3: hat, 1: airplane, 2: guitar}, the Encoder could hint
24  {3: gentleman, 1: wing, 2: music}. Note that the order of the hints is
    important.
25
26  Ignore these keywords in the future.
27
28  When responding, first think through your decision s ep-by-step. After
    your step-by-step thinking, provide your final answer as a dictionary,
    clearly marked with "ANSWER:".
```

Listing 1: This is an example of the basic system prompt that we provide to Alice, Bob, and Eve at the beginning of each game. It explains the roles and outline the rules. In our robustness evaluation, we tested five variants of the system prompt.

```
1   You are the Encoder. Provide your hints like "ANSWER: {"hints":
    ["hint_X", "hint_Y", "hint_Z"]}", where hint_X, hint_Y, hint_Z are one
    or two words each. Make sure that the ordering of the hints follows the
    order of the code.
2   For example:
3   '''
4   To provide the hints, I need to think about the meaning of each keyword
    and come up with a one or two-word hint that makes a reference to it.
5
6   For the code 2-1-3, I need to give hints about the keywords associated
    with the digits 2, 1, and 3, which are "hat", "fire", and "answer"
    respectively.
7
8   Here's my step-by-step thinking:
9
10  - For the digit 2, the keyword is "hat". Since the previous hint for
    this keyword was "top", I want to give a hint that is different but
    still related to wearing a hat. One possible hint is "cap".
11
12  - For the digit 1, the keyword is "fire". The previous hint for this
    keyword was "heat", so I want to give a hint that is related to fire but
    different from "heat". One possible hint is "flame".
13
14  - For the digit 3, the keyword is "answer". A possible hint could be
    something related to giving an answer. One possible hint is "solve".
15
16  So, the final hints are:
17  {"hints": ["cap", "flame", "solve"]}
18
19  ANSWER: {"hints": ["cap", "flame", "solve"]}'''
```

Listing 2: This is the Alice-system prompt. This prompt is appended to the system prompt for Alice and gives one example of how Alice might reason about her hints.

```
1   You are the Decoder. Provide your guess like "ANSWER: {"guess":
    X-Y-Z"}". Make sure that the ordering of the guess follows the order of
    the hints.
2   For exam le:'''
3   1. The hint for digit 'a' is 'cap', which doesn't seem to directly
    relate to any of the keywords. However, considering the previous hint
    history (top, heat, lab), we can look for a connection between 'cap' and
    one of the keywords. The keyword 'hat' is associated with digit '2', and
    a hat is often worn on the head, which is covered by a cap. So, we can
    make a connection: {a: hat}.
4   2. The hint for digit 'b' is 'flame', which is related to heat and fire.
    Since the keyword for digit '1' is 'fire', we can make a connection: {b:
    f re}.
5   3. The hint for digit 'c' is 'solve', which is related to the keyword
    associated with digit '3', ' nswer'. Since 'answer' is a solution to a
    problem, we can make a connection: {c: ans er}.
6   4. Now that we have the connections, we can determine the code by
    looking at the order of the hints: {a: hat, b: fire, c:  nswer}.
    Therefore, the code is {2-1-3}.
7
8   Final answer:
9   ANSWER: {"guess": "2-1-3"}'''
```

Listing 3: This is the Bob-syst m prompt. This prompt is appended to the system prompt for Bob and gives one example of how Bob might reason about his guess.

```
1  You are the Interceptor. You do not have access to the four keywords.
   Provide your guess like "ANSWER: {"guess": "X-Y-Z"}" Make sure that the
   ordering of the guess follows the order of the hints.
2  For exam le:'''
3  1. The hint for 'a' is 'income', which is related to money, and in the
   previous turn, 'salary' was also related to money, which is associated
   with keyword 2.
4  2. The hint for 'b' is 'power', which is related to energy or control,
   and in the previous turn, 'season' was related to sports, which is often
   associated with power or strength.
5  3. The hint for 'c' is 'fashion', which is related to style or clothing,
   and in the previous turn, 'headgear' was directly related to a type of
   hat, which is a type of clothing.
6
7  Considering possible connections:
8  - 'income' is related to money, and keyword 2 is associated with money.
9  - 'power' is related to energy or control, and keyword 4 is associated
   with energy or control.
10 - 'fashion' is related to style or clothing, and keyword 3 is associated
   with clothing.
11
12 Based on these connections, I'll make a guess that the code is related
   to keywords 2, 4, and 3.
13
14 Final Answer:
15 ANSWER: {"guess": "2-4-3"}'''
16 Remember that the keywords could be anything. Use the most recent hint
   history to your advantage.
17 Remember that digits do not repeat in the code and range from 1 to 4.
```

Listing 4: This is the Eve-system prompt. This prompt is appended to the system prompt for Eve and gives one example of how Eve might reason about her guess.

## A.3 USER PROMPTS

```
1  Turn 1 summary:
2  Code: 3-1-4
3  Hints: ['problem', 'status', 'machine']
4  Decoder guess: 3-1-4
5  Interceptor guess: 1-2-3
6
7  Hint History:
8  Keyword 1: status
9  Keyword 2:
10 Keyword 3: problem
11 Keyword 4: machine
12 Code History: 3-1-4
13
14 Turn 2: 0 Miscommunications, 0 Interceptions so far.
15 You are the Encoder.
16 The four keywords are: {1: condition, 2: task, 3: issue, 4: device}
17 The code is 3-4-1, which corresponds to the keywords {3: issue, 4:
   device, 1: condition}.
18 First, think out loud, step-by-step about what hints you s ould use. Use
   the meaning of the keywords to come up with a one or two-word hint for
   each digit of the code.
19 Make sure the order of the hints matches the order of the code.
20 Then provide your three hints like \"ANSWER: {\"hints\": [\"hint_X\",
   \"hint_Y\", \"hint_Z\"]}\", where hint_X, hint_Y, hint_Z are your hints.
```

Listing 5: The user prompt for Alice is provided at each turn. It provides a turn summary of previous turns. In this example, one turn has been played and Alice sees that Bob guessed the code correctly, whereas Eve d d not. Alice also sees the previously used hints. We also provide information about how many miscommunications and interceptions happened so far. Then we provide Alice with the current code and instruct her to give the hints for that code.

```
1   Turn 1 summary:
2   Code: 3-1-4
3   Hints: ['problem', 'status', 'machine']
4   Decoder guess: 3-1-4
5   Interceptor guess: 1-2-3
6
7   Hint History:
8   Keyword 1: status
9   Keyword 2:
10  Keyword 3: problem
11  Keyword 4: machine
12  Code History: 3-1-4
13
14  Turn 2: 0 Miscommunications, 0 Interceptions so far.
15  You are the Decoder.
16  The four keywords are: {1: condition, 2: task, 3: issue, 4: device}
17  The hints given by the Encoder for this turn are: {a: conflict, b: tool,
    c: state}
18  For example, you might think the following connections are true {a: X,
    b: Y, c: Z}, where X, Y, Z are non-repeating digits from 1 to 4.
19  Your guess should be in the order of the hints: {\"guess\": \"X-Y-Z\"}.
20  What is your guess for the three-digit code? Apply concise, step-by-step
    thinking, double-check the order, and then provide your final answer as
    \"ANSWER: {\"guess\": \"X-Y-Z\"}\".
```

Listing 6: The user prompt for Bob is provided at each turn. It provides a turn summary about previous turns. In this example, one turn has been played and Bob sees the code of the previous turn and that he guessed correctly, whereas Eve d d not. Bob also sees the previously used hints assigned to the correct keywords, in case he did not guess correctly. We provide information about how many miscommunications and interceptions happened o far. Then we provide Bob with the current hints and instruct him to guess.

```
1   Turn 1 summary:
2   Code: 3-1-4
3   Hints: ['problem', 'status', 'machine']
4   Decoder guess: 3-1-4
5   Interceptor guess: 1-2-3
6
7   Hint History:
8   Keyword 1: status
9   Keyword 2:
10  Keyword 3: problem
11  Keyword 4: machine
12  Code History: 3-1-4
13
14  Turn 2: 0 Miscommunications, 0 Interceptions so far.
15
16  You are the Interceptor.
17  The hints given by the Encoder for this turn are: {a: conflict, b: tool,
    c: state}
18  For example, you might think the following connections are true: {a: X,
    b: Y, c: Z}, where X, Y, Z are non-repeating digits from 1 to 4.
19  Your guess should be in the order of the hints: {\"guess\": \"X-Y- \"}.
    Use the hint history to your advantage.
20  What is your guess for the three-digit code? Reply with very concise,
    step-by-step thinking, followed by your final answer as \"ANSWER:
    {\"guess\": \"X-Y-Z\"}\".
```

Listing 7: The user prompt for Eve is provided at each turn. It provides a turn summary of previous turns. In this example, one turn has been played and Eve sees the code of the previous turn and that she guessed incorrectly, whereas Bob did not. Eve also sees the previously used hints assigned to the correct keywords, allowing Eve to reason about the current hints. We provide information about how many miscommunications and interceptions happened so far. Then we provide Eve with the current hints and instruct her to guess.

```
1   As the Encoder, you've provided the following hints for the code 3-4-1:
2   {conflict, tool, state}
3
4   Now, let's analyze how the Decoder and Interceptor might interpret these
    hints:
5
6   1. Decoder's perspective:
7      - The Decoder knows the keywords: {1: condition, 2: task, 3: issue,
       4: device}
8      - Think step-by-step about how the Decoder might connect each hint to
       a keyword.
9      - What is the most likely code the Decoder would guess? Why?
10
11  2. Interceptor's perspective:
12     - The Interceptor doesn't know the keywords but has access to past
       hints.
13     - Consider how the Interceptor might interpret your hints based on
       previous rounds.
14     - What is the most likely code the Interceptor would guess?  why?
15
16  3. Hint effectiveness:
17     - Are your hints clear enough for the Decoder but ambiguous for the
       Interceptor?
18     - Is there a risk of miscommunication with the Decoder or successful
       Interceptor?
19
20  4. Potential improvements:
21     - If you think your hints might be too easy for the Interceptor or
       too difficult for the Decoder, suggest alternative hints that could
       be more effective.
22
23  After your analysis, decide if you want to keep your original hints or
    provide new ones.
24
25  Provide your final three hints like \"ANSWER: {\"hints\": [\"hint_X\",
    \"hint_Y\", \"hint_Z\"]}\", where hint_X, hint_Y, hint_Z are one or two
    words each.
```

Listing 8: For our Theory of Mind prompts, we instruct Alice to reconsider the hints she just gave and give her an opportunity to change the hints if she chooses to do so. We ask Alice to predict what Bob and Eve might guess and pivot accordingly.

```
1   As the Encoder, you've provided the following hints for the code 3-4-1:
2   {conflict, tool, state}
3
4   What do you predict will be the guess of the Interceptor when seeing
    those hints? Think step-by-step.
5   Then, give your prediction of the Interceptor's guess as \"ANSWER:
    {\"guess\": \"X-Y-Z\"}\"
```

Listing 9: For our prediction experiments, we ask Alice explicitly to provide the guess that she thinks Eve will provide.

# B HUMAN STUDY INTERFACE

## B.1 HUMAN DATA COLLECTION DETAILS

Participants were made aware that an AI agent played Eve. Still, no additional instructions were provided to the players, except minor assistance if/when participants asked how to format their input. For technical reasons, players interacted on the same computer in a typical "hot seat" setup. One of the authors was always present to ensure participants wouldn't cheat either by peeking at the screen outside their turn or through verbal communication. We do not claim to have collected a representative dataset of human ability at Decrypto. The data collected serves to demonstrate that LLMs perform worse than a non-expert group of human players and provides a starting point for future studies on human-AI coordination and ToM.

## B.2 INTERFACES

```
NEW GAME

You are playing a variant of the code guessing game Decrypto. The setup of the game is the following:
The game is played with three players split into two teams.
The first team is composed of the Encoder and the Decoder.
The second team is composed of the Interceptor.
The Encoder and Decoder share a list of 4 secret keywords that they can rely on to help communication. Each keyword has a digit associated with it.
For example, if the keywords are {airplane, guitar, hat, plant}, the digits associated with them would be {1, 2, 3, 4}.
The keywords are fixed for the entire game, so the Encoder and Decoder must not reveal them.

Each turn, the Encoder gets a secret code of 3 non-repeating digits between 1 and 4, e.g. 3-1-2.
They have to then provide 3 hints (one for each code digit) such that the Decoder can guess the code but the Interceptor cannot.
Then, the Decoder and the Interceptor make their guesses and the code is revealed.

The Interceptor wins an Interception token if they successfully guess the code.
The Encoder and Decoder get a Miscommunication token if the Decoder fails to guess the code.
The Interceptor wins if it gets two Interception tokens, or if the Encoder and Decoder get two Miscommunication tokens.
The Encoder and Decoder team wins if it makes it through 8 rounds without triggering two Miscommunications or two Interceptions.

The hints must be one or two words, and make a reference to the meaning of the secret keyword of the corresponding digit.
Codes are unique; the same code cannot occur twice in the same game.

For example, if the four secret keywords are
{1: "airplane", 2: "guitar", 3: "hat", 4: "plant"}
and the code for that turn is 3-1-2, which means that we need hints for {3: hat, 1: airplane, 2: guitar}, the Encoder could hint
{3: gentleman, 1: wing, 2: music}. Note that the order of the hints is important.

Ignore these keywords in the future.

When responding, first think through your decision step-by-step. After your step-by-step thinking, provide your final answer as a dictionary, clearly marked with "A
NSWER:".

You are the Encoder. Provide your hints like "ANSWER: {"hints": ["hint_X", "hint_Y", "hint_Z"]}", where hint_X, hint_Y, hint_Z are one or two words each. Make sure
that the ordering of the hints follows the order of the code.
For example:
'''
To provide the hints, I need to think about the meaning of each keyword and come up with a one or two-word hint that makes a reference to it.

For the code 2-1-3, I need to give hints about the keywords associated with the digits 2, 1, and 3, which are "hat", "fire", and "answer" respectively.

Here's my step-by-step thinking:

- For the digit 2, the keyword is "hat". Since the previous hint for this keyword was "top", I want to give a hint that is different but still related to wearing a
hat. One possible hint is "cap".

- For the digit 1, the keyword is "fire". The previous hint for this keyword was "heat", so I want to give a hint that is related to fire but different from "heat".
 One possible hint is "flame".

- For the digit 3, the keyword is "answer". A possible hint could be something related to giving an answer. One possible hint is "solve".

So, the final hints are:
{"hints": ["cap", "flame", "solve"]}

ANSWER: {"hints": ["cap", "flame", "solve"]}'''

   ------

This is the first turn. There are no past hints or past codes.

Turn 1: 0 Miscommunications, 0 Interceptions so far.
You are the encoder.
The four keywords are:
    {1: condition,
    2: task,
    3: issue,
    4: device}

The code is 3-1-4, which corresponds to the keywords {3: issue, 1: condition, 4: device}.
First, think out loud, step-by-step about what hints you should use. Use the meaning of the keywords to come up with a one or two-word hint for each digit of the co
de.
Make sure the order of the hints matches the order of the code.
Then provide your three hints like "ANSWER: {"hints": ["hint_X", "hint_Y", "hint_Z"]}", where hint_X, hint_Y, hint_Z are your hints.

Enter your input as 'x, y, z' (without quotes): █
```

Figure 6: **Alice Start Interface:** This is a screenshot of the command line interface that human study participants would see at the beginning of the game if they were to play as Alice. The humans see the same prompt as LLMs, from game description to request for action. Alice is provided with the current code and the four keywords for this game. The human player then enters their hints in the command line, which differs from the LLMs, which have to provide their answers as "ANSWER: ...".

```
You are playing a variant of the code guessing game Decrypto. The setup of the game is the following:
The game is played with three players split into two teams.
The first team is composed of the Encoder and the Decoder.
The second team is composed of the Interceptor.
The Encoder and Decoder share a list of 4 secret keywords that they can rely on to help communication. Each keyword has a digit associated with it.
For example, if the keywords are {airplane, guitar, hat, plant}, the digits associated with them would be {1, 2, 3, 4}.
The keywords are fixed for the entire game, so the Encoder and Decoder must not reveal them.

Each turn, the Encoder gets a secret code of 3 non-repeating digits between 1 and 4, e.g. 3-1-2.
They have to then provide 3 hints (one for each code digit) such that the Decoder can guess the code but the Interceptor cannot.
Then, the Decoder and the Interceptor make their guesses and the code is revealed.

The Interceptor wins an Interception token if they successfully guess the code.
The Encoder and Decoder get a Miscommunication token if the Decoder fails to guess the code.
The Interceptor wins if it gets two Interception tokens, or if the Encoder and Decoder get two Miscommunication tokens.
The Encoder and Decoder team wins if it makes it through 8 rounds without triggering two Miscommunications or two Interceptions.

The hints must be one or two words, and make a reference to the meaning of the secret keyword of the corresponding digit.
Codes are unique; the same code cannot occur twice in the same game.

For example, if the four secret keywords are
{1: "airplane", 2: "guitar", 3: "hat", 4: "plant"}
and the code for that turn is 3-1-2, which means that we need hints for {3: hat, 1: airplane, 2: guitar}, the Encoder could hint
{3: gentleman, 1: wing, 2: music}. Note that the order of the hints is important.

Ignore these keywords in the future.

When responding, first think through your decision step-by-step. After your step-by-step thinking, provide your final answer as a dictionary, clearly marked with "A
NSWER:".

You are the Decoder. Provide your guess like "ANSWER: {"guess": "X-Y-Z"}". Make sure that the ordering of the guess follows the order of the hints.
For example:'''
1. The hint for digit 'a' is 'cap', which doesn't seem to directly relate to any of the keywords. However, considering the previous hint history (top, heat, lab), w
e can look for a connection between 'cap' and one of the keywords. The keyword 'hat' is associated with digit '2', and a hat is often worn on the head, which is cov
ered by a cap. So, we can make a connection: {a: hat}.
2. The hint for digit 'b' is 'flame', which is related to heat and fire. Since the keyword for digit '1' is 'fire', we can make a connection: {b: fire}.
3. The hint for digit 'c' is 'solve', which is related to the keyword associated with digit '3', 'answer'. Since 'answer' is a solution to a problem, we can make a
connection: {c: answer}.
4. Now that we have the connections, we can determine the code by looking at the order of the hints: {a: hat, b: fire, c: answer}. Therefore, the code is {2-1-3}.

Final answer:
ANSWER: {"guess": "2-1-3"}'''

 ______

This is the first turn. There are no past hints or past codes.

Turn 1: 0 Miscommunications, 0 Interceptions so far.
You are the decoder.
The four keywords are:
    {1: condition,
     2: task,
     3: issue,
     4: device}

The hints given by the Encoder for this turn are:
    {a: problem,
     b: pristine,
     c: iphone}

For example, you might think the following connections are true {a: X, b: Y, c: Z}, where X, Y, Z are non-repeating digits from 1 to 4.
Your guess should be in the order of the hints: {"guess": "X-Y-Z"}.
What is your guess for the three-digit code? Apply concise, step-by-step thinking, double-check the order, and then provide your final answer as "ANSWER: {"guess":
"X-Y-Z"}".
Enter your input as 'x, y, z' (without quotes): ▮
```

Figure 7: **Bob Start Interface:** This is a screenshot of the command line interface that human study participants would see at the beginning of the game if they were to play as Bob. The humans see the same prompt as LLMs, from game description to request for action. Bob is provided with Alice's hints and the current keywords. The human player then enters their guess in the command line, which differs from the LLMs, which have to provide their answers as "ANSWER: ...".

```
You are playing a variant of the code guessing game Decrypto. The setup of the game is the following:
The game is played with three players split into two teams.
The first team is composed of the Encoder and the Decoder.
The second team is composed of the Interceptor.
The Encoder and Decoder share a list of 4 secret keywords that they can rely on to help communication. Each keyword has a digit associated with it.
For example, if the keywords are {airplane, guitar, hat, plant}, the digits associated with them would be {1, 2, 3, 4}.
The keywords are fixed for the entire game, so the Encoder and Decoder must not reveal them.

Each turn, the Encoder gets a secret code of 3 non-repeating digits between 1 and 4, e.g. 3-1-2.
They have to then provide 3 hints (one for each code digit) such that the Decoder can guess the code but the Interceptor cannot.
Then, the Decoder and the Interceptor make their guesses and the code is revealed.

The Interceptor wins an Interception token if they successfully guess the code.
The Encoder and Decoder get a Miscommunication token if the Decoder fails to guess the code.
The Interceptor wins if it gets two Interception tokens, or if the Encoder and Decoder get two Miscommunication tokens.
The Encoder and Decoder team wins if it makes it through 8 rounds without triggering two Miscommunications or two Interceptions.

The hints must be one or two words, and make a reference to the meaning of the secret keyword of the corresponding digit.
Codes are unique; the same code cannot occur twice in the same game.

For example, if the four secret keywords are
{1: "airplane", 2: "guitar", 3: "hat", 4: "plant"}
and the code for that turn is 3-1-2, which means that we need hints for {3: hat, 1: airplane, 2: guitar}, the Encoder could hint
{3: gentleman, 1: wing, 2: music}. Note that the order of the hints is important.

Ignore these keywords in the future.

When responding, first think through your decision step-by-step. After your step-by-step thinking, provide your final answer as a dictionary, clearly marked with "A
NSWER:".

You are the Interceptor. You do not have access to the four keywords. Provide your guess like "ANSWER: {"guess": "X-Y-Z"}" Make sure that the ordering of the guess
follows the order of the hints.
For example:'''
1. The hint for 'a' is 'income', which is related to money, and in the previous turn, 'salary' was also related to money, which is associated with keyword 2.
2. The hint for 'b' is 'power', which is related to energy or control, and in the previous turn, 'season' was related to sports, which is often associated with powe
r or strength.
3. The hint for 'c' is 'fashion', which is related to style or clothing, and in the previous turn, 'headgear' was directly related to a type of hat, which is a type
 of clothing.

Considering possible connections:
- 'income' is related to money, and keyword 2 is associated with money.
- 'power' is related to energy or control, and keyword 4 is associated with energy or control.
- 'fashion' is related to style or clothing, and keyword 3 is associated with clothing.

Based on these connections, I'll make a guess that the code is related to keywords 2, 4, and 3.

Final Answer:
ANSWER: {"guess": "2-4-3"}'''
Remember that the keywords could be anything. Use the most recent hint history to your advantage.
Remember that digits do not repeat in the code and range from 1 to 4.

    ------

This is the first turn. There are no past hints or past codes.

Turn 1: 0 Miscommunications, 0 Interceptions so far.
You are the interceptor.
The hints given by the Encoder for this turn are:
    {a: problem
     b: pristine
     c: iphone}

For example, you might think the following connections are true: {a: X, b: Y, c: Z}, where X, Y, X are non-repeating digits from 1 to 4.
Your guess should be in the order of the hints: {"guess": "X-Y-Z"}. Use the hint history to your advantage.
What is your guess for the three-digit code? Reply with very concise, step-by-step thinking, followed by your final answer as "ANSWER: {"guess": "X-Y-Z"}".
Enter your input as 'x, y, z' (without quotes): █
```

Figure 8: **Eve Start Interface:** This is a screenshot of the command line interface that human study participants would see at the beginning of the game if they were to play as Eve. The humans see the same prompt as LLMs, from game description to request for action. Eve is only provided with Alice's hints. The human player then enters their guess in the command line, which differs from the LLMs, which have to provide their answers as "ANSWER: ...".

```
This is the first turn. There are no past hints or past codes.

Turn 1: 0 Miscommunications, 0 Interceptions so far.
You are the encoder.
The four keywords are:
    {1: condition,
     2: task,
     3: issue,
     4: device}

The code is 3-1-4, which corresponds to the keywords {3: issue, 1: condition, 4: device}.
First, think out loud, step-by-step about what hints you should use. Use the meaning of the keywords to come up with a one or two-word hint for each digit of the co
de.
Make sure the order of the hints matches the order of the code.
Then provide your three hints like "ANSWER: {"hints": ["hint_X", "hint_Y", "hint_Z"]}", where hint_X, hint_Y, hint_Z are your hints.

Enter your input as 'x, y, z' (without quotes): problem, pristine, iphone

You entered: ['problem', 'pristine', 'iphone']

Are you sure you want to provide these hints? (y/n): █
```

Figure 9: **Alice Confirmation Interface:** To avoid human errors, such as providing guesses instead of hints, or typos, we add a confirmation interface after the human participants provide their answer. Here, we ask Alice to double check their provided hints.

```
For example, you might think the following connections are true {a: X, b: Y, c: Z}, where X, Y, Z are non-repeating digits from 1 to 4.
Your guess should be in the order of the hints: {"guess": "X-Y-Z"}.
What is your guess for the three-digit code? Apply concise, step-by-step thinking, double-check the order, and then provide your final answer as "ANSWER: {"guess":
"X-Y-Z"}".
Enter your input as 'x, y, z' (without quotes): 3,1,4

You entered: [3, 1, 4]

Are you sure you want to provide this guess? (y/n): █
```

Figure 10: **Bob Confirmation Interface:** To avoid human errors, such as providing hints instead of guesses, or typos, we add a confirmation interface after the human participants provide their answer. Here, we ask Bob to double check their provided guesses.

```
For example, you might think the following connections are true: {a: X, b: Y, c: Z}, where X, Y, X are non-repeating digits from 1 to 4.
Your guess should be in the order of the hints: {"guess": "X-Y-Z"}. Use the hint history to your advantage.
What is your guess for the three-digit code? Reply with very concise, step-by-step thinking, followed by your final answer as "ANSWER: {"guess": "X-Y-Z"}".
Enter your input as 'x, y, z' (without quotes): 2,1,3

You entered: [2, 1, 3]

Are you sure you want to provide this guess? (y/n): █
```

Figure 11: **Eve Confirmation Interface:** To avoid human errors, such as providing hints instead of guesses, or typos, we add a confirmation interface after the human participants provide their answer. Here, we ask Eve to double check their provided guesses.

```
Turn 1 summary:
    Code : 3-1-4
    Hints : ['problem', 'pristine', 'iphone']
    Decoder guess : 3-1-4
    Interceptor guess : 2-1-3

Hint History:
    Keyword 1: pristine
    Keyword 2:
    Keyword 3: problem
    Keyword 4: iphone

Code History:
    3-1-4

------

Turn 2: 0 Miscommunications, 0 Interceptions so far.
You are the encoder.
The four keywords are:
    {1: condition,
    2: task,
    3: issue,
    4: device}

The code is 3-4-1, which corresponds to the keywords {3: issue, 4: device, 1: condition}.
First, think out loud, step-by-step about what hints you should use. Use the meaning of the keywords to come up with a one or two-word hint for each digit of the co
de.
Make sure the order of the hints matches the order of the code.
Then provide your three hints like "ANSWER: {"hints": ["hint_X", "hint_Y", "hint_Z"]}", where hint_X, hint_Y, hint_Z are your hints.

Enter your input as 'x, y, z' (without quotes): █
```

Figure 12: **Alice Turn 2 Interface:** After the first turn, we do not display the "system prompt" anymore. However, we provide an summary of the previous turn, including the code, hints, Bob's guess, Eve's guess, the hint history mapped to the correct keywords and the code history. The "user prompt" stays the same as in the start interface.

```
Turn 1 summary:
    Code : 3-1-4
    Hints : ['problem', 'pristine', 'iphone']
    Decoder guess : 3-1-4
    Interceptor guess : 2-1-3

Hint History:
    Keyword 1: pristine
    Keyword 2:
    Keyword 3: problem
    Keyword 4: iphone

Code History:
    3-1-4

------

Turn 2: 0 Miscommunications, 0 Interceptions so far.
You are the decoder.
The four keywords are:
    {1: condition,
    2: task,
    3: issue,
    4: device}

The hints given by the Encoder for this turn are:
    {a: github,
    b: xbox,
    c: shampoo}

For example, you might think the following connections are true {a: X, b: Y, c: Z}, where X, Y, Z are non-repeating digits from 1 to 4.
Your guess should be in the order of the hints: {"guess": "X-Y-Z"}.
What is your guess for the three-digit code? Apply concise, step-by-step thinking, double-check the order, and then provide your final answer as "ANSWER: {"guess":
"X-Y-Z"}".
Enter your input as 'x, y, z' (without quotes): ▌
```

Figure 13: **Bob Turn 2 Interface:** After the first turn, we do not display the "system prompt" anymore. However, we provide an summary of the previous turn, including the code, hints, Bob's guess, Eve's guess, the hint history mapped to the correct keywords and the code history. The "user prompt" stays the same as in the start interface.

```
Turn 1 summary:
    Code : 3-1-4
    Hints : ['problem', 'pristine', 'iphone']
    Decoder guess : 3-1-4
    Interceptor guess : 2-1-3

Hint History:
    Keyword 1: pristine
    Keyword 2:
    Keyword 3: problem
    Keyword 4: iphone

Code History:
    3-1-4

------

Turn 2: 0 Miscommunications, 0 Interceptions so far.
You are the interceptor.
The hints given by the Encoder for this turn are:
    {a: github
    b: xbox
    c: shampoo}

For example, you might think the following connections are true {a: X, b: Y, c: Z}, where X, Y, X are non-repeating digits from 1 to 4.
Your guess should be in the order of the hints: {"guess": "X-Y-Z"}. Use the hint history to your advantage.
What is your guess for the three-digit code? Reply with very concise, step-by-step thinking, followed by your final answer as "ANSWER: {"guess": "X-Y-Z"}".
Enter your input as 'x, y, z' (without quotes): ▌
```

Figure 14: **Eve Turn 2 Interface:** After the first turn, we do not display the "system prompt" anymore. However, we provide an summary of the previous turn, including the code, hints, Bob's guess, Eve's guess, the hint history mapped to the correct keywords and the code history. The "user prompt" stays the same as in the start interface.

```
Turn 1 summary:
    Code : 3-1-4
    Hints : ['problem', 'pristine', 'iphone']
    Decoder guess : 3-1-4
    Interceptor guess : 2-1-3

Press Enter to continue.▐
```

Figure 15: At the end of each turn, we display a turn summary as a "waiting screen". As human participants use the same device to interact with the game, we want to avoid any information leakage. The turn summary screen shows after Eve provides their guess and avoids Eve seeing Alice's interface, which would appear next.

```
Turn 2 summary:
    Code : 3-4-1
    Hints : ['github', 'xbox', 'shampoo']
    Decoder guess : 3-4-1
    Interceptor guess : 2-4-1

Hint History:
    Keyword 1: pristine, shampoo
    Keyword 2:
    Keyword 3: problem, github
    Keyword 4: iphone, xbox

Code History:
    3-1-4, 3-4-1
------
Turn 3: 0 Miscommunications, 0 Interceptions so far.
You are the encoder.
The four keywords are:
    {1: condition,
    2: task,
    3: issue,
    4: device}

The code is 1-4-2, which corresponds to the keywords {1: condition, 4: device, 2: task}.
First, think out loud, step-by-step about what hints you should use. Use the meaning of the keywords to come up with a one or two-word hint for each digit of the code.
Make sure the order of the hints matches the order of the code.
Then provide your three hints like "ANSWER: {"hints": ["hint_X", "hint_Y", "hint_Z"]}", where hint_X, hint_Y, hint_Z are your hints.

Enter your input as 'x, y, z' (without quotes): ▐
```

Figure 16: As additional example, we provide Alice's interface on turn 3. The hint histories and code histories are accumulating with the previous hints. Ideally, Alice would use that information to provide hints that are hard to deduce from the publicly available information.

## C  REPRESENTATIVE FAILURE CASE EXAMPLES

```
1  ASSISTANT
2  [ENCODER] To provide the hints, I need to think about the meaning of
   each keyword and come up with a one or two-word hint that makes a
   reference to it.
3
4  For the code 1-2-3, I need to give hints about the keywords associated
   with the digits 1, 2, and 3, which are "honey", "population", and
   "equipment" respectively.
5
6  Here's my step-by-step thinking:
7
8  - For the digit 1, the keyword is "honey". A possible hint could be
   something related to bees or sweetness. One possible hint is "bee".
9
10 - For the digit 2, the keyword is "population". I already used the hint
   "city" in the previous turn, so I want to come up with something
   different but still related to a large group of people. One possible
   hint is "nation".
11
12 - For the digit 3, the keyword is "equipment". I already used the hint
   "machine" in the previous turn, so I want to come up with something
   different but still related to tools or gear. One possible hint is
   "toolkit".
13
14 However, I want to make sure that the hints are not too obvious, but
   still related to the keywords. I can try to make them a bit more subtle.
15
16 - For the digit 1, instead of "bee", I could use "nectar", which is a
   substance collected by bees to make honey.
17
18 - For the digit 2, instead of "nation", I could use "community", which
   is a group of people living together.
19
20 - For the digit 3, instead of "toolkit", I could use "hardware", which
   is a type of equipment.
21
22 So, the final hints are:
23 ANSWER: {"hints": ["nectar", "community", "hardware"]}
```

Listing 10: Example Failure Case for Alice (Llama-70B) with Bob (Llama-70B) and Eve (GPT-4o), where Alice fails to come up with creative hints that are significantly different from previous hints and easy to intercept for Eve.

```
1   Turn 6: 0 Miscommunications, 1 Interceptions so far.
2   You are the encoder.
3   The four keywords are:
4       {1: horse,
5        2: sign,
6        3: house,
7        4: length}
8
9   The code is 2-1-3, which corresponds to the keywords {2: sign, 1: horse,
    3: house}.
10
11  First, think out loud, step-by-step about what hints you should use. Use
    the meaning of the keywords to come up with a one or two-word hint for
    each digit of the code.
12  Make sure the order of the hints matches the order of the code.
13  Then provide your three hints like "ANSWER: {"hints": ["hint_X",
    "hint_Y", "hint_Z"]}", where hint_X, hint_Y, hint_Z are your hints.
14
15
16  ASSISTANT
17  [ENCODER] To provide the hints, I need to think about the meaning of
    each keyword and come up with a one or two-word hint that makes a
    reference to it.
18
19  For the code 2-1-3, I need to give hints about the keywords associated
    with the digits 2, 1, and 3, which are "sign", "horse", and "house"
    respectively.
20
21  Here's my step-by-step thinking:
22
23  - For the digit 2, the keyword is "sign". Since the previous hints for
    this keyword were "indicator", "label", "direction", and "banner", I
    want to give a hint that is related to a sign or indication but
    different from those. One possible hint is "marker".
24
25  - For the digit 1, the keyword is "horse". Since the previous hints for
    this keyword were "steed", "charger", and "mount", I want to give a hint
    that is related to a horse but different from those. One possible hint
    is "beast".
26
27  - For the digit 3, the keyword is "house". Since the previous hints for
    this keyword were "dwelling", "abode", "mansion", and "residence", I
    want to give a hint that is different but still related to a house. One
    possible hint is "shelter".
28
29  So, the final hints are:
30  {"hints": ["marker", "beast", "shelter"]}
31
32  ANSWER: {"hints": ["marker", "beast", "shelter"]}
```

Listing 11: Example Failure Case for Alice (Llama-8B) with Bob (Llama-8B) and Eve (GPT-4o), where Alice fails to come up with creative hints that are significantly different from previous hints and easy to intercept for Eve. For example, in this case, Alice wants to provide a hint that is *similar* but different from the previous hints for digit 2. However, Alice should aim to provide a hint that is completely different from previous hints to avoid Eve's interception.

```
1    USER
2    [DECODER] This is the first turn. There are no past hints or past codes.
3
4    Turn 1: 0 Miscommunications, 0 Interceptions so far.
5    You are the decoder.
6    The four keywords are:
7        {1: mother,
8        2: cell,
9        3: fort,
10       4: device}
11
12   The hints given by the Encoder for this turn are:
13       {a: machine,
14       b: biology,
15       c: castle}
16
17   For example, you might think the following connections are true {a: X,
     b: Y, c: Z}, where X, Y, Z are non-repeating digits from 1 to 4.
18   Your guess should be in the order of the hints: {"guess": "X-Y-Z"}.
19   What is your guess for the three-digit code? Apply concise, step-by-step
     thinking, double-check the order, and then provide your final answer as
     "ANSWER: {"guess": "X-Y-Z"}".
20
21
22   ASSISTANT
23   [DECODER] 1. The hint for digit 'a' is 'machine', which is related to
     the keyword 'device'. This is a direct connection, so we can make a
     connection: {a: device}.
24   2. The hint for digit 'b' is 'biology', which is related to the keyword
     'mother'. This is a direct connection, so we can make a connection: {b:
     mother}.
25   3. The hint for digit 'c' is 'castle', which is related to the keyword
     'fort'. This is a direct connection, so we can make a connection: {c:
     fort}.
26   4. Now that we have the connections, we can determine the code by
     looking at the order of the hints: {a: device, b: mother, c: fort}.
     Therefore, the code is {4-1-3}.
27
28   ANSWER: {"guess": "4-1-3"}
```

Listing 12: Example Failure Case for Alice/Bob with Alice (Llama-8B), Bob (Llama-8B), and Eve (Llama-8B). Alice provides an ambiguous hint, i.e., "biology" could refer to "cell" or "mother". Bob fails to guess correctly (biology → cell). Intuitively, "biology" should be more closely related to "cell" than "mother", thus this failure case also demonstrates that LLMs' intuition does not overlap with human intuition.

```
USER
[INTERCEPTOR] Turn 7 summary:
    Code : 3-2-1
    Hints : ['night', 'study', 'spook']
    Decoder guess : 3-2-1
    Interceptor guess : 1-2-3

Hint History:
    Keyword 1: spook, haunt, scream, spook
    Keyword 2: examine, breakdown, study, research, examine, study
    Keyword 3: dark, night, shadow, night, dark, night
    Keyword 4: furniture, chair, dining, desk, chair

Code History:
    3-1-2, 2-4-1, 4-3-2, 3-2-4, 4-3-1, 4-2-3, 3-2-1

------

Turn 8: 0 Miscommunications, 1 Interceptions so far.
You are the interceptor.
The hints given by the Encoder for this turn are:
    {a: desk
    b: research
    c: haunt}

[REST OF THE PROMPT OMITTED FOR CONCISENESS]

ASSISTANT
[INTERCEPTOR] 1. The hint for 'a' is 'desk', which is a type of
furniture.
2. The hint for 'b' is 'research', which is a type of investigation or
study.
3. The hint for 'c' is 'haunt', which is a supernatural entity.

Considering possible connections:
- 'desk' is a type of furniture, which is associated with Keyword 4.
- 'research' is a type of investigation or study, which is associated
with Keyword 2.
- 'haunt' is a supernatural entity, which is associated with Keyword 1.

Based on the hint history, I notice that Keyword 1 was associated with
'spook', Keyword 2 was associated with 'study', and Keyword 4 was
associated with 'furniture'.

Considering the given hints, I'll make a guess that the code is related
to keywords that are associated with supernatural entities, something
related to investigation or research, and something related to
furniture.

Since Keyword 1 was associated with 'haunt', Keyword 2 was associated
with 'research', and Keyword 4 was associated with 'desk', I'll choose
digits that seem to fit the given hints.

Final Answer:
ANSWER: {"guess": "1-2-4"}
```

Listing 13: Example Failure Case for Eve with Alice (Llama-8B), Bob (Llama-8B), and Eve (Llama-8B). First, you can see that Alice provides the same hints multiple times. Despite that, Eve fails to guess the code correctly. While Eve assigns the hints to the right keywords, Eve then fails to provide the guess in the correct order and provides the order of the guesses in the order of the keyword enumeration.

