# OpenReview forum: "The Decrypto Benchmark for Multi-Agent Reasoning and Theory of Mind"
_ICLR.cc/2025/Conference — Submitted to ICLR 2025_

### Official Review · Reviewer_9Qt8 · 2024-11-03

**Soundness:** 3
**Presentation:** 4
**Contribution:** 4
**Rating:** 8
**Confidence:** 4

**Summary:**

The authors propose the Decrypto benchmark, a novel interactive evaluation framework for testing multi-agent coordination, competition, and ToM capabilities in AI systems.  Decrypto a popular (and very fun!) boardgame requires agents to reason about other agents' knowledge and decision-making while engaging in both cooperative and competitive gameplay. The benchmark is designed to be future-proof for large language models (LLMs) while avoiding common pitfalls of existing benchmarks like data leakage and lack of interactivity.

**Strengths:**

- I think it's fantastic and increasingly fruitful when studies leverage approaches from cognitive science to study AI, and in particular when they use boardgames to test (human and) AI capabilities because it provides a controlled environment, but one where you can arbitrarily scale complexity as needed. Decrypto in particular is a great game for testing cooperation, ToM, adherence to RSA, etc. (and is also a great game in general, though no bonus review points for that :-) ). Kelsey Allen et al. beautifully lay out the case for these kinds of studies in their NHB paper "Using games to understand the mind"
- The authors make a compelling case for using games as benchmarks, particularly for evaluating theory of mind capabilities, by drawing clear connections to foundational cognitive science work like the Three Mountain Problem
- The benchmark design thoughtfully eliminates common LLM failure modes (like numerical computation and tokenization issues) to focus specifically on reasoning and coordination abilities
- The empirical evaluation is comprehensive, including ablation studies, human-AI cross-play experiments, and detailed analysis of model performance across different roles
-The authors demonstrate that even state-of-the-art LLMs struggle with this benchmark despite its focus on language-based reasoning, revealing important limitations in current systems

**Weaknesses:**

- The theoretical analysis of the benchmark's properties could be strengthened, particularly regarding what specific aspects of theory of mind and coordination are being tested. In particular, this may benefit from leaning on the Rational Speech Act literature
- The authors could expand on how the benchmark's difficulty scales with agent capabilities; while they mention it cannot be saturated, more formal analysis would be valuable
 - The human baseline data collection (9 games) seems limited given the importance of human comparisons in the results section. Crucially, I also don't see important details about the human data collection in the manuscript (what were the demographics of the human players, what was their prior experience with Decrypto, were they compensated for their participation, was the study cleared by IRB, what was the interface that humans saw, etc.)
- The authors could provide more detailed analysis of failure modes; what specific types of reasoning or coordination break down when LLMs perform poorly?
 - A discussion of potential gaming or adversarial strategies would strengthen the benchmark's robustness claims

*Note: I think the paper is great and would be very happy to re-evaluate my score if the authors address my concerns.*

**Questions:**

- How do the authors ensure that the keyword corpus provides consistent difficulty across different games?
- Were any metrics considered beyond win rates and game length for evaluating performance?
- Could the authors elaborate on how the benchmark tests different levels of ToM reasoning?
- How sensitive is agent performance to the specific choice of prompts and system messages?
- See weaknesses above for some additional questions

**Details Of Ethics Concerns:**

Not sure if ethics flag is needed or not, but I don't see important details about the human subjects in the manuscript including whether the study received IRB approval, how participants were chosen, how they were compensated, etc.

---

> ### Author Response · Authors · 2024-11-21
> **Authors Response (1/3)**
>
> We thank the reviewer for their constructive review and for raising a few weaknesses we omitted. Their feedback has helped us improve the paper significantly. We are also glad that they recognize the value (and fun!) of using Decrypto as a tool to evaluate reasoning, cooperation and ToM.
>
> > _The theoretical analysis [...] could be strengthened, particularly regarding what specific aspects of theory of mind and coordination are being tested. In particular, this may benefit from leaning on the Rational Speech Act literature_
>
> We attempted a theoretical analysis of Decrypto in the RSA framework, following the definition in [1]. Decrypto can be seen as two coupled iterated implicature games: one between Alice and Bob and another between Alice and Eve. In both, the meaning space includes all possible codes and the utterance space all possible triples of hints provided by Alice.  Agents have uniform priors over the codes not yet sampled on previous turns.
>
> However, Degen clearly states three limitations of RSA: ``It is typically applied in small toy meaning domains with very limited sets of utterance alternatives, requires explicitly specifying the literal semantics of all utterance alternatives, and does not extend beyond single-shot utterance production/interpretation.''
>
> In contrast, Decrypto has an utterance space at least the size of the English dictionary cubed. The literal semantics of all utterances are very hard to specify, because Alice can explicitly exploit multi-hop associations and cultural references going beyond literal interpretations of a word. Finally, the game is inherently multi-shot, with interpretations evolving at each round.
>
> That said, Decrypto requires agents to model the beliefs and knowledge of all other players, and, in the case of Alice, to make strategic decisions based on that model. In addition, we now explicitly measure 3 different ToM abilities (up from 1 in the original submission). The original one is perspective understanding, similar to Piaget's Three Body Problem. We now also include an experiment measuring Representational Change and False Belief, as defined by Gopnik & Astington [2].
>
> The reviewer can find our new experiment in Section 5 -- Theory of Mind and in Table 3 of the revised paper. The takeaway is that while ToM abilities correlate with model size, both Llama 3.1 8B and 70B struggle with more refined representational change and modeling of false beliefs, with scores particularly low ($\leq 17\\%$ pass rate) for what we consider the "Strong" variants of the tasks.
>
> > _The authors could expand on how the benchmark's difficulty scales with agent capabilities; while they mention it cannot be saturated, more formal analysis would be valuable_
>
> We are unsure how to draw a formal argument for why difficulty scales with agent capabilities. If the reviewer has intuitions on how to conduct such an analysis, we would love them to tell us.
>
> That said, Decrypto is a 2-team zero-sum game where the action space for one agent (Alice) is entirely open-ended, since Alice can select hints from the entirety of the English language. Given this, the optimal strategy is likely impossible to compute, and depends on the knowledge and reasoning abilities of each of the models in the game. Stronger Encoders can use more subtle hints, by relying on uncommon definitions or engaging in multi-hop reasoning, while stronger Decoders and Interceptors become better at identifying relationships between words.
>
> Our results support this idea, since we empirically show that models which are generally considered more powerful perform better in all roles. We also demonstrate in Figure 2 that it is possible to craft an arbitrarily strong Encoder-Decoder team, but that the members of such a team may fail to coordinate with other agents Difficulty can also arise from knowledge disparities between models, which further enriches the benchmark. Indeed, if Alice and Bob have different knowledge cutoffs (or one is human while the other is an LLM), the players must be cognizant of each other's limitations.
>
> To further reduce saturation, we have 680 potential keywords, resulting in over 8.8 billion combinations. This makes it nearly impossible to expose the model to more than a minuscule fraction of possible games. The number of combinations matters because keyword combinations determine whether individual hints are good or not. For instance, "vacation" can be a clear hint for "summer", but becomes ambiguous if another keyword is "beach".
>
> Nonetheless, we have adjusted the wording in the paper to add nuance to our claims on saturation.

---

> ### Author Response · Authors · 2024-11-21
> **Authors Response (2/3)**
>
> > _Limited human data and missing human data collection details_
>
> We appreciate the reviewer's concerns regarding human data collection. The data collection and release was approved following the standard review process at our institution for this kind of studies, with the condition of not collecting any identifying information about the players, including demographics.
>
> The 9 games were collected from 5 total adult participants. Only one of the participants had prior experience with Decrypto and no compensation was provided for participation.
>
> Players interacted with the game through a command-line interface operated on the author's computer. They received the same prompt (i.e., instructions) as we provided the LLMs, which were formatted in the same way. This includes the rules, which are part of the system prompt. Screenshots of the interface are now in Appendix B. Participants were made aware that an AI agent played Eve. No additional instructions were provided to the players, except minor assistance if/when participants asked how to format their input. For technical reasons, players interacted on the same computer in a typical "hot seat" setup. One of the authors was always present to ensure participants wouldn't cheat either by peeking at the screen outside their turn or through verbal communication.
>
> We highlight that we do not claim to have collected a representative dataset of human ability at Decrypto. The data collected demonstrates that LLMs perform worse than a non-expert group of human players and provides a starting point for future studies on human-AI coordination and ToM, which we clarified in the updated version of the text.
>
> We have now added these details to the paper. We also commit to increase the number of total games and participants we include in the study for the final version of the paper. We will also add Human interceptor results to Table 1, since we realize they are missing.
>
> Finally, we will open-source all human game data alongside the benchmark code.
>
> > _provide more detailed analysis of failure modes; what specific types of reasoning or coordination break down when LLMs perform poorly?_
>
> We thank the reviewer for raising this point.
>
> At a high level, Alice and Bob can lose either due to miscommunications or to interceptions, and we study both in Figure 3. The former condition corresponds to a coordination failure between Alice and Bob, but only occurs in at most 16% of games with LLM teams (when (Alice, Bob) = (GPT-4o, Llama-8B) and as little as 0% for stronger teams. Meanwhile, interceptions make up no less than 71% and up to 100% of end game conditions when all three players are LLMs.
>
> This indicates the main failure mode is that Alice provides hints that are too obvious. This is something we also confirm by inspecting the game logs, where we find that all hints for a keyword will generally be very semantically aligned. For instance, for the keyword "fire", Alice might hint something like "flame", "heat" and "spark".
>
> We also show in Table 2 that Alice struggles to predict Eve's guess, even when both are played by the same LLM. There is no indication that Alice even attempts to reason about Eve's guess unprompted when selecting hints, but if she did, her predictions would often fail.
>
> We updated the manuscript to provide examples of representative failure cases in Appendix C.
>
> > _discussion of potential gaming or adversarial strategies_
>
> We provide two hard-coded baselines and show in Figure 2 that by relying on a shared (and arbitrary) notion of word similarity, it is possible to create a virtually unbeatable Encoder-Decoder team. As such, creating a team of specialist agents to "solve" Decrypto is not interesting -- we basically did that already.
>
> Where the challenge arises is in creating a generalist agent capable of playing with or against other generalist agents, including humans. Approaches here require creativity to come up with new hints, ToM, and actively modeling the knowledge available to other agents. Because hints are open-ended, the space of strategies is enormous. For instance, if Alice and Bob have specialized knowledge or a later knowledge cutoff than Eve, they could exploit that by basing their hints on recent events or jargon specific to their expertise.
>
> This also goes to show that, while Eve currently has a much higher win rate in our experiments, her role is not objectively easier. In principle, Alice can make references to the entirety of human culture and human public knowledge, with an arbitrary number of reasoning steps. For Eve to ever intercept the codes, she needs knowledge and reasoning abilities that are comparable to Alice's.

---

> ### Author Response · Authors · 2024-11-21
> **Authors Response (3/3)**
>
> **Questions**
>
> > _How do the authors ensure that the keyword corpus provides consistent difficulty across different games?_
>
> Decrypto does not require difficulty to be consistent across different games. To ensure representative results, most of our results are reported over 32 game instances and 3 model seeds. Some keyword combinations are harder than others. That said, difficulty is also highly subjective, depending on the players' knowledge. For instance, "plane" is much easier to hint at if Alice and Bob know all 4 definitions of the word (flat surface, airplane, woodworking tool, type of tree).
>
> > _metrics considered beyond win rates and game length_
>
> Yes, we use intercepts and/or miscommunications in figures 2, Figure 3 and Table 1 to provide more insight and granularity into how agents perform. The number of miscommunications measures Alice’s and Bob’s ability to cooperate, and the number of interceptions captures Alice and Eve’s ability to compete. There is a tension between miscommunications and interceptions in Decrypto. Alice can aim to minimise miscommunications and provide easy hints. If Alice’s hints were easy to guess, Alice would never miscommunicate with Bob, but Eve could certainly intercept the code. If Alice aims to minimise intercepts, the hints would be impossible to guess for Eve, but also for Bob. Thus, the number of miscommunications and intercepts are two sides of the same coin. Average game length per game allows us to capture both sides in one metric. If the average length is high, Alice and Bob can successfully balance the difficulty of hints to avoid miscommunications and intercepts.
>
> > _Could the authors elaborate on how the benchmark tests different levels of ToM reasoning?_
>
> We are unsure what the reviewer means by ``different levels'' of ToM. Can they please clarify? In particular, we welcome any treatise of ToM levels of reasoning that would help us analyze the game and agent performance.
>
> Otherwise, we point the reviewer to our ToM experiments, discussed earlier in our response.
>
> > _How sensitive is agent performance to the specific choice of prompts and system messages?_
>
> Not very sensitive. We evaluate this in Figure 4 with 625 prompt variations (including system prompts) and found that model size is the biggest determining factor of ability in Decrypto. We also found that the smaller model is more sensitive to prompt variation than the larger one tested. We updated the manuscript to highlight the robustness study in Section 5 -- Robustness.
>
> We thank the reviewer for their great feedback! We hope to have addressed their concerns and if so, that they will consider updating their score. We also hope to one day play Decrypto with them.
>
> [1] Judith Degen, The Rational Speech Act Framework, 2023
>
> [2] Gopnik & Astington, Children's Understanding of Representational Change and Its Relation to the Understanding of False Belief and the Appearance-Reality Distinction

---

> > ### Author Response · Authors · 2024-11-27
> > **A Decrypto Puzzle**
> >
> > As the manuscript revision period comes to end, we ask whether the reviewer has had a chance to consider our response, our "General Response to all Reviewers", and our updated manuscript. Based on the reviewers' feedback, we have **improved and clarified multiple key points in the paper**, and **included a new experiment measuring two additional ToM abilities** following references provided by reviewer AGZj.
> >
> > If they wish, we also invite the reviewer to try and solve a Decrypto puzzle:
> >
> > ```
> > You play as Eve. Analyse hints from previous rounds and the current hints to intercept the code. For this puzzle, assume all codes of 3 non-repeating digits are possible.
> >
> > Hints from previous rounds:
> > 1. patience, moral, knight
> > 2. money, luck, wheel
> > 3. dog, blue, choke
> > 4. water, oil, pearl
> >
> > Hints from this round:
> > a. neck
> > b. vice
> > c. value
> >
> > What is the code?
> > ```
> >
> > Finally, if we have addressed the reviewer's concerns, we kindly ask them to update their score.

---

> > > ### Author Response · Authors · 2024-12-02
> > >
> > > Dear reviewer 9Qt8,
> > >
> > > Today is the final day of the discussion period. As such, we kindly ask the reviewer if they have any additional feedback following our response. We also ask that they consider increasing their support for our paper if their concerns have been addressed.
> > >
> > > Thank you,
> > >
> > > The Decrypto Authors

---

> ### Comment · Reviewer_9Qt8 · 2024-12-03
>
> I very deeply appreciate the authors carefully engaging with my feedback in their responses.
> I apologize for the last minute reply, but I'm very happy to increase my score for this submission and hope to see it at the conference!

---

### Official Review · Reviewer_AGZj · 2024-11-04

**Soundness:** 2
**Presentation:** 1
**Contribution:** 2
**Rating:** 5
**Confidence:** 4

**Summary:**

The paper proposes to use a game Decrypto, to evaluate coordination, competition, and theory of mind (ToM) reasoning capabilities in agentic, foundational AI models. They show experiments on LLMs, and human-AI cross-play.

**Strengths:**

Decrypto introduces a unique  framework for studying language understanding and communication. Unlike traditional benchmarks that rely on static inputs, Decrypto involves dynamic clue-based exchanges where the meaning evolves over time. This setup offers a more naturalistic challenge for AI models, better reflecting the way humans adapt language in interactive and often ambiguous contexts.

**Weaknesses:**

Looking at the design of the task and its inspiration from Piagets three mountains task , I think this paper falls short as a theory of mind benchmark. The original Piagets three mountains task, primarily tests spatial perspective taking. Likewise in Decrypto, the task only requires players to think about shifting between "clues" or coded language without attributing beliefs or knowledge to others. Since Eve and Alice know each others codes at the end of the round, it could become more of a pattern matching game rather than ToM. Can you please clarify this. The claim is central to the paper.

Also, Decrypto does not seem to require players to reason about or infer others' beliefs—particularly beliefs that could be mistaken or based on incomplete information. I mean, in Decrypto, players decode and interpret clues, but if the gameplay primarily revolves around interpreting words or phrases without attributing or understanding others' mental states (e.g., “they think that we think X”), it does not measure ToM in the way Gopnik and Astington define. Without this, Decrypto would align more closely with tasks focused on communication or strategy rather than Theory of Mind.


References :

Flavell, J. H. (1992). Perspectives on perspective-taking
Gopniks Children's Understanding of Representational Change and Its Relation to the Understanding of False Belief and the Appearance-Reality Distinction
Wellman, H. M., Cross, D., & Watson, J. (2001). Meta-analysis of theory-of-mind development: The truth about false belief

**Questions:**

Comments
The paper is a bit hard to read, please try to improve writing
Table 1 is not referred anywhere in the paper!
Couldn't find ablation even though it is referred in the paper
Abstract ends abruptly!
L83: Decrypto isolates language-based reasoning and association, directly leveraging LLMs’ core training objective. Where is this explained?
line 191 values of GPT-4o in the left column are doesn't end, incomplete  sentence
There is no baseline where Humans are intercepted by Humans to know how difficult or easy is this task. I think this is important for the claim about the task evaluating ToM.

---

> ### Author Response · Authors · 2024-11-21
> **Authors Response (1/2)**
>
> We thank the reviewer for their thorough review, and are glad they saw the value of Decrypto benchmark. We also thank them for the provided references, and in particular Gopnik 1988, which helped us improve the paper significantly.
>
> > _I think this paper falls short as a theory of mind benchmark [...] the task only requires players to think about shifting between "clues" or coded language without attributing beliefs or knowledge to others_
>
> We firmly believe Decrypto requires extensive ToM. Although codes are revealed at the end of each turn, keywords are never revealed to Eve. Furthermore, hints are not selected from a fixed vocabulary. They can be any word or phrase and can rely on any form of public knowledge, including technical knowledge or popular culture. We illustrate with an example why Alice requires the attribution of beliefs or knowledge to others to play Decrypto well.
>
> Suppose, after the turn shown in Figure 1, that Alice picks up the code 4-2-3. What would be a good hint for digit 4 (plane)? Eve already has the hints {wing, turbulence, takeoff} for Keyword 4, so it is reasonable to attribute to Eve the belief that "Keyword 4 is related to aviation or flight." If Alice fails to do so, she may make another direct reference to an airplane or aviation and likely get intercepted.
>
> Instead, Alice may choose the hint "bark", which refers to "plane trees", i.e. trees in the genus Platanus, which have a distinctive bark. This is, however, only a good hint if Alice a) believes that Bob has enough botanical knowledge to think of "plane" as a type of tree and b) believes that Bob believes that Alice would make this connection as well (e.g. as opposed to something like bark -> trees -> giraffes).
>
> This shows two ways in which Alice reasoned about the knowledge of the other players. She reasoned about the in-game knowledge available to Eve by considering that Eve would believe digit 4 to be related to aviation and reasoned about Bob's real-world knowledge by assuming he would know that "planes" are a type of tree.
>
> Other agents must also similarly reason about each other's beliefs. For instance, Alice might choose more or less obvious hints depending on whether her team accumulated miscommunication or interception tokens so far, which in turn can influence how Bob and Eve should interpret the hints.
>
> Furthermore, the word-embeddings baseline performance indicates that perfect theory of mind results in strong performance. If a baseline Alice/Bob pair share the same word embeddings, i.e., they have a accurate model of the others word associations, they play arbitrarily well, as Alice can perfectly predict what Bob will guess. We clarified this connection in the updated manuscript.
>
> In short, while Decrypto "gameplay primarily revolves around interpreting words or phrases", the correct interpretation is intrinsically linked to maintaining beliefs about other agents' knowledge and mental state.

---

> ### Author Response · Authors · 2024-11-21
> **Authors Response (2/2)**
>
> > _[Decrypto] does not measure ToM in the way Gopnik and Astington define_
>
> We thank the reviewer for this insightful reference. First, we would like to highlight that there is an analogy between our experiment in Table 2 and the false-belief problem in Gopnik and Astington. In that paper, children see a box of smarties and its content (pencils) and must predict what another child would guess as being the content of the box given only a view of the closed box. Similarly, Alice sees the hints provided and the keywords, and must predict what another player (Eve) would guess as being the keywords given only the hints.
>
> However, the paper inspired us to conduct an additional ToM experiment, which explicitly measures representational change (RC) and false belief (FB) abilities in the way presented by Gopnik and Astington. We describe this experiment here, and also in Section 5 -- Theory of Mind of the revised manuscript.
>
> Focusing on Eve, we branch out the context of the agent and prompt it three times independently. The first prompt asks it to predict the four keywords. The second prompt reveals the keywords and asks the model what it thought were the keywords before the reveal. The third prompt again reveals the keywords and asks the model to predict what a ``second Interceptor'' who has seen everything except the reveal would think are the keywords.
>
> By comparing the first and the second answers, we measure RC, the ability of the agent to recognize when its belief about the world (but not the world itself) changes due to additional information. By comparing the first and third answers, we measure FB, the ability to represent other agents as having inaccurate beliefs about the world. For the Strong variant of those tasks, we consider the agent to pass if it correctly predicts what it answered in question 1. We consider an agent correct for the Weak variant if the answers to questions 2 or 3 are not the real keywords. Results in Table 3 show that ability correlates with model size but that neither of the models gets perfect scores. On Strong tasks, pass rates are particularly low ($\leq 17\\%$ pass rate), evidence that LLMs only do not have persistent models of their ``mind'' of that of others.
>
> > _please try to improve writing_
>
> We have now improved writing throughout the paper, including all points raised by the reviewer. We will further improve writing for the camera ready version.
>
> > _baseline where Humans are intercepted by Humans_
>
> Due to time constraints of the rebuttal, we are not able to report these results within the allotted period. However, we will add these results for the camera-ready version.
>
> > _Couldn't find ablation_
>
> We perform an ablation over prompts in Section 6 -- Robustness, and present the results in Figure 4. We now rewrote part of the abstract and the paper to clarify this.
>
> > _"Decrypto isolates language-based reasoning and association, directly leveraging LLMs’ core training objective." Where is this explained?_
>
> We explain how Decrypto isolates language abilities in Section 3 (l. 134-142). In particular, we aim to propose a benchmark that is as "easy" as possible for LLMs by eliminating the need for skills such as tool use, symbolic, mathematical or spatial reasoning, long-term planning or careful tokenization.
>
> Instead, the benchmark is operated entirely through word associations, something that LLMs could reasonably be expected to excel at, since learning word co-occurrences is an important step towards reducing perplexity during pre-training. We now clarify this in the paper.
>
> We once again thank the reviewer, as their feedback genuinely helped us improve the paper. We hope to have addressed the reviewer's concerns and if so that they will consider improving their support for our paper.

---

> > ### Author Response · Authors · 2024-11-27
> > **A Decrypto Puzzle**
> >
> > As the manuscript revision period comes to end, we ask whether the reviewer has had a chance to consider our response, our "General Response to all Reviewers", and our updated manuscript. Based on the reviewers' feedback, we have **improved and clarified multiple key points in the paper**, and **included a new experiment measuring two additional ToM abilities** following references provided by reviewer AGZj.
> >
> > If they wish, we also invite the reviewer to try and solve a Decrypto puzzle:
> >
> > ```
> > You play as Eve. Analyse hints from previous rounds and the current hints to intercept the code. For this puzzle, assume all codes of 3 non-repeating digits are possible.
> >
> > Hints from previous rounds:
> > 1. patience, moral, knight
> > 2. money, luck, wheel
> > 3. dog, blue, choke
> > 4. water, oil, pearl
> >
> > Hints from this round:
> > a. neck
> > b. vice
> > c. value
> >
> > What is the code?
> > ```
> >
> > Finally, if we have addressed the reviewer's concerns, we kindly ask them to update their score.

---

> > > ### Author Response · Authors · 2024-12-02
> > > **Final day of discussion**
> > >
> > > Dear reviewer AGZj,
> > >
> > > Today is the final day of the discussion period. As such, we kindly ask the reviewer if they have any additional feedback following our response. We also ask that they consider increasing their support for our paper if their concerns have been addressed.
> > >
> > > Thank you,
> > >
> > > The Decrypto Authors

---

> > > > ### Comment · Reviewer_AGZj · 2024-12-03
> > > >
> > > > Sorry for the delayed reply.
> > > > But some of my concerns with the paper remain.
> > > >
> > > > I agree that having TOM does help the task but follow up experiment does not fully address the concern, as this could be solved with finding a right encoding decoding schema between Alice and Bob. Though I see that there is no explicit communication channel between Alice and Bob, the latter just being able to do better pattern matching can still help in the task. I am not sure if you answered this question in the rebuttal. Sorry if I missed.
> > > >
> > > > Consider the proposed experiment being done over multiple runs, with the same agents in the respective roles (say 10 games with 8 turns each). And if we see how the performance improves over games, then I think the claim that the models indeed creates a model of the 'beliefs’ of another model holds.
> > > >
> > > > I would like to clarify your experiment where different LLMs are used. Performance seems to vary based on the sophistication of the Eve model. When a stronger model acts as Eve, the number of interceptions increases, indicating that the ability of eve is dependent on its general capability.
> > > > On the other hand if the task was measuring TOM in isolation. One would expect high performance in scenarios where: Alice and Bob are simpler yet the same model (they actually have same representations) and Eve is a powerful yet different model.
> > > >
> > > > I didn't see the results pointing in this direction. Hopefully I didn't get this wrong.
> > > > Also on studying representation differences in LLM, I would be more convinced if you could show something with the actual representation of LLM rather than using prompt based proxy tasks (which is the only resort when it comes to understanding representation in humans).
> > > >
> > > > In short if the benchmark in measuring ToM
> > > > Eve performance (LLM1,LLM1,LLM2) <  Eve performance (LLM1,LLM1,LLM1)
> > > > Given LLM1~LLM2 in general capabilities. Given the same LLMs unlike humans will have the exact same representation. Also prior work shows some ability to corporate.

---

> > > > > ### Author Response · Authors · 2024-12-04
> > > > > **Follow-up by the Authors**
> > > > >
> > > > > > "this could be solved with finding a right encoding decoding schema"
> > > > >
> > > > > Not only "could"; we did solve Decrypto with our baselines (see L. 271). Results in Figure 2 show that if Alice and Bob share a representation and encoding-decoding strategy, they can perform arbitrarily well. But the representations do not generalize: for large enough K, they are not intercompatible nor are they compatible with LLMs (Fig. 2) or humans (Table 1). They also break as soon as they encounter words not included in the original Word2Vec or GloVe vocabulary.
> > > > >
> > > > > The challenge arises in creating a generalist agent capable of playing with or against other generalist agents, including humans. Approaches here require creativity to come up with new hints, ToM, and actively modeling the knowledge available to other agents. Because hints are open-ended, the space of strategies is enormous. For instance, if Alice and Bob have specialized knowledge or a later knowledge cutoff than Eve, they could exploit that by basing their hints on recent events or jargon specific to their expertise.
> > > > >
> > > > > We refer the reviewer back to the "Authors Response (1/2)" above, and in particular to the discussion surrounding the hint "bark" for an example. Whether "bark" is a good hint for "plane" is a matter of the beliefs Alice and Bob hold of each other's botanical knowledge.
> > > > >
> > > > > We discuss the distinction between specialist agents (purpose-built for Decrypto) and generalist agents at length in Section 4.1.
> > > > >
> > > > > > "if we see how the performance improves over games, then I think the claim that the models indeed creates a model of the 'beliefs’ of another model holds."
> > > > >
> > > > > That is correct, and is what happens between human players. If humans play together repeatedly (or know each other), their ability to coordinate in Decrypto is enhanced. For instance, in a casual game of Decrypto, one of the authors used the word "push" to hint for the keyword "star". This worked because they knew that Bob was also a researcher and would likely associate both "push" and "star" as being Github operations. They also knew that Eve had limited programming knowledge, and was therefore unlikely to make this link.
> > > > >
> > > > > To be clear, at we do not claim that LLMs actively maintain beliefs of other agents. In fact, **our own ToM experiments provide strong evidence that agents do not engage in ToM and do not actively model other agents**. As such, we expect that playing multiple games in a row would do little to improve LLM performance, unlike for humans.
> > > > >
> > > > > > "if you could show something with the actual representation of LLM"
> > > > >
> > > > > We already provide results with our baselines that are tied to their underlying representations (GloVe or Word2Vec). Studying LLM performance in Decrypto using mechanistic interpretability is a very promising avenue, but beyond the scope of this paper.
> > > > >
> > > > > > "if the benchmark in measuring ToM Eve performance (LLM1,LLM1,LLM2) < Eve performance (LLM1,LLM1,LLM1) Given LLM1~LLM2."
> > > > >
> > > > > That absolutely correct. We see precisely this phenomenon in Figure 3, on the bottom right, which reports Avg. Turns per Episode for Alice=Bob and Eve chosen separately. Looking specifically at baselines:
> > > > >
> > > > > - (Word2Vec, Word2Vec, Word2Vec) : 4.34 turns
> > > > > - (Word2Vec, Word2Vec, GloVe) : 4.59 turns
> > > > > - (GloVe, GloVe, GloVe) : 4.25 turns
> > > > > - (GloVe, GloVe, Word2Vec) : 4.45 turns
> > > > >
> > > > > Shorter episodes mean that Eve intercepts faster, and so in both cases Eve's performance is higher if Eve = Alice/Bob, exactly as predicted by the reviewer. And GloVe ~ Word2Vec since they have the same underlying algorithm and similar performance in Fig. 2. The only difference is their underlying representation, as the reviewer pointed out.
> > > > >
> > > > > Such an effect does not appear for LLMs because **they are not good enough at Decrypto**. The hints given by LLM Alice are too obvious, in that they are close to the keyword and past hints in most "reasonable" representations. For instance, if the keyword is "fire", Alice might provide the hints {flame, heat, smoke} instead of, e.g. {pottery, gun, chestnuts} (as in "firing pottery", "gunfire" and "roasting chestnuts on an open fire"). Then, even a weak Eve is likely to associate "smoke" to "flame" and "heat" and is likely to intercept the code, regardless of whether it is the same LLM as Alice or not.
> > > > >
> > > > > We hope the results obtained with our baselines and the three examples we mentioned (bark/plane, push/star and fire) demonstrate how ToM is crucial for high performance in Decrypto, and how agents with shared knowledge or representations can coordinate much better as Alice and Bob, as long as they actively exploit that shared representation.
> > > > >
> > > > > We hope our response addresses the Reviewer's remaining concerns. We will also strive to further improve writing, and to clarify what high-level play looks like in Decrypto, even though current LLMs are far from reaching that point. This will be on top of the multiple writing improvements we already brought to the paper, highlighted in blue in the revised PDF.

---

### Official Review · Reviewer_Z9ie · 2024-11-06

**Soundness:** 2
**Presentation:** 3
**Contribution:** 2
**Rating:** 5
**Confidence:** 4

**Summary:**

The paper introduces Decrypto, an interactive benchmark for evaluating multi-agent reasoning and theory of mind capabilities in foundational AI models. It involves a language-based board game to test coordination, competition, and strategic reasoning. Results show that large language models (LLMs) struggle to coordinate with both humans and other LLMs, indicating significant limitations in theory of mind capabilities compared to humans.

**Strengths:**

1. The paper introduces a novel benchmark that specifically targets theory of mind capabilities, providing a unique approach to multi-agent reasoning evaluation.

2. It presents a benchmark designed to assess the performance of LLMs in both cooperative and competitive multi-agent environments, which is significant for advancing the field. The benchmark isolates language-based reasoning, offering a focused evaluation of LLMs without requiring complex symbolic or spatial reasoning, enhancing clarity.

3. The proposed benchmark allows for adaptability in difficulty levels, addressing saturation issues in other benchmarks and ensuring scalability for future models.

**Weaknesses:**

1. The method lacks rigor in controlling for potential confounding variables, which casts doubt on the validity of the observed differences in model performance across experimental settings.

2. The experimental design and descriptions lack precision, particularly in detailing how varying LLM architectures impact theory of mind capabilities, leaving key assumptions and decisions unaddressed.

3. Evaluation metrics appear inadequate, as they rely heavily on average turn length without addressing the potential noise introduced by differing prompt conditions, which may skew the results.

4. The ablation studies are limited in scope and fail to explore the interaction between vocabulary size and hint similarity thoroughly, leaving significant gaps in understanding model robustness. And the comparative analysis with baseline models is poorly justified, as it assumes without evidence that the chosen baselines are representative of the broader capabilities of specialist versus generalist agents.

**Questions:**

1. In the methodology section, could you clarify how you account for confounding variables when comparing the theory of mind performance across different LLM architectures?

2. Your evaluation relies significantly on average turn length as a performance metric. Could you elaborate on the rationale behind choosing this metric and explain how you mitigate potential noise introduced by prompt variations?

3. The scope of the ablation studies appears limited, particularly regarding the interaction between vocabulary size and hint similarity. Could you provide more detailed experiments that explore how these factors interact and influence model robustness?

4. The selection of baseline models to represent “specialist” versus “generalist” agents seems insufficiently justified. How did you determine that these baselines are representative of broader capabilities within these categories?

**Details Of Ethics Concerns:**

No.

---

> ### Author Response · Authors · 2024-11-21
> **Authors Response (1/2)**
>
> We thank the reviewer for their comments. We are happy they recognize Decrypto as a "novel benchmark" "isolating language-based reasoning" and that "specifically targets Theory of Mind". We are also glad that they agree that competitive and cooperative evaluation, something Decrypto enables, is "significant for advancing the field".
>
> > _The method lacks rigor in controlling for potential confounding variables_
>
> We have done our best to rule out confounding variables and evaluate LLMs according to the best standards in the field. In particular, we run an ablation on our choice of prompts, reporting the score over 625 different prompt combinations covering Alice and Bob, as well as system and instruction prompts. The reviewer may find this study in Section 5 -- Robustness and in Figure 4, where we show prompts have a minor effect on performance compared to model size. We now also provide additional details on how the prompts were constructed in the revised manuscript. This prompt robustness study vastly exceeds prompt studies for other benchmarks, if such a study was conducted in the first place. [1][2][3][4][5][6]
>
> We also report our results aggregated over 32 episodes and 3 model seeds to account for the varying difficulty of keyword combinations and sampling variance in generative models. This alone distinguishes us from well-established LLM benchmarks [7,8] who seldom report uncertainty. If the reviewer believes we have omitted an important confounding variable, we ask that they please tell us which.
>
> > _The experimental design and descriptions lack precision, particularly in detailing how varying LLM architectures impact theory of mind capabilities, leaving key assumptions and decisions unaddressed._
>
> We study Theory of Mind (ToM) capabilities in two different ways. First, as the reviewer themselves highlight, Decrypto is a "benchmark that specifically targets theory of mind capabilities", since reasoning about the beliefs and knowledge available to other players is essential for performing well in the game. As such, we provide extensive results for open and closed-source models ranging from Llama 3.1 8B to GPT-4o, looking at both self- and cross-play across the different roles.
>
> Second, we set up an explicit ToM experiment in Table 2, and that ToM capabilities do not necessarily increase with model size.
>
> While Section 3 (Theory of Mind) and Section 5 (Theory of Mind) already details the ToM experiments and results, we further highlighted the impact of the results in the updated version of the paper.
>
> We kindly ask the reviewer to specify which key assumptions and decisions they believe we left unaddressed.
>
> > _Evaluation metrics appear inadequate, as they rely heavily on average turn length without addressing the potential noise introduced by differing prompt conditions_
>
> We respectfully disagree. We explicitly study the impact of different prompts in Section 5 -- Robustness, where we evaluate 625 different prompt combinations. We find that prompts matter less than model size for predicting performance.
>
> In addition to game length, we report miscommunications (Figure 2, 3 and Table 1), intercepts (Figure 3, Table 1) and win rate (Figure 2). Nonetheless, we chose to report average game length as it encapsulates both the competitive and cooperative aspect of Decrypto while being more granular than win rate, which we clarified in the updated manuscript.
>
> > _the comparative analysis with baseline models is poorly justified, as it assumes without evidence that the chosen baselines are representative of the broader capabilities of specialist versus generalist agents._
>
> We appreciate the reviewer's concern and now include additional justification for the design choices behind our baselines in Section 4.1. Note that we never claimed our baselines to be representative of specialist agents. In fact, we explicitly discuss other possible types of specialist agents in the original submission, including ``rule-based strategies'' and large pre-trained models pre-trained on game data.
>
> However, our baselines remain representative of at least a subset of possible specialist agents, since we can tune them to perform arbitrarily well by changing $K$, as shown in Figure 2. In doing so, we show a trade-off between self-play and cross-play performance. We also establish an upper bound on performance to which to compare other agents, including generalist ones.

---

> ### Author Response · Authors · 2024-11-21
> **Authors Response (2/2)**
>
> > _evaluation relies significantly on average turn length [,,,] Could you elaborate on the rationale behind choosing this metric and explain how you mitigate potential noise introduced by prompt variations?_
>
> We thank the reviewer for raising this concern. While we already motivated our choice in Section 3 (Competition), we updated the manuscript to further explain the average turn length choice.
>
> The rationale is as follows. The key metrics provided naturally by the game are the number of interceptions, the number of miscommunications and certainly the number of wins either by Alice/Bob or Eve. The number of miscommunications measures the ability of Alice and Bob to cooperate and the number of interceptions captures Eve’s ability to compete. There is a tension between miscommunications and interceptions in Decrypto. Alice can aim to minimise miscommunications and provide easy hints. If Alice’s hints were easy to guess, Alice would never miscommunicate with Bob, but Eve could certainly intercept the code. If Alice aims to minimise intercepts, the hints would be impossible to guess for Eve, but also for Bob. Thus, the number of miscommunications and intercepts are two sides of the same coin. Average game length per game allows us to capture both sides in one metric. If the average length is high, Alice and Bob can successfully balance the difficulty of hints to avoid miscommunications and intercepts.
>
> As stated above, we provide a robustness study in Figure 4 and analysis thereof in Section 5 (Robustness).
>
> > _The scope of the ablation studies appears limited, particularly regarding the interaction between vocabulary size and hint similarity. Could you provide more detailed experiments that explore how these factors interact and influence model robustness?_
>
> There is no fixed hint corpus for the LLM-based agents. The LLM-based agents may choose any hint that their tokenisation allows. The hint corpus is only used for the word-embedding-based baselines. Word embeddings are not generative models like LLMs and thus cannot generate arbitrary hints. The baselines select the hints based on word similarity (measured by cosine similarity) between the current keywords and all the possible hints in the hint corpus. The baseline then selects a hint randomly from the TopK most similar words. Figure 2 shows the interaction between vocabulary size (size of K) and the hint similarity as measured in Miscommunication rates. As hints become less similar to keywords, the number of miscommunications increases.
> We hope this clarifies the distinction between baselines with a fixed vocabulary size and LLM agents.
>
> We’d like to thank the reviewer again for their review and questions, which helped us improve the clarity of the paper. We have now uploaded a revised version of the paper, which, along with our response, hopefully addresses the reviewer's concerns. If so, we ask the reviewer to kindly increase their support for our paper.
>
> [1]  Zhuang Chen, Jincenzi Wu, Jinfeng Zhou, Bosi Wen, Guanqun Bi, Gongyao Jiang, Yaru Cao,
> Mengting Hu, Yunghwei Lai, Zexuan Xiong, et al. Tombench: Benchmarking theory of mind
> in large language models. arXiv preprint arXiv:2402.15052, 2024b
>
> [2] Anthony Costarelli, Mat Allen, Roman Hauksson, Grace Sodunke, Suhas Hariharan, Carlson Cheng,
> Wenjie Li, and Arjun Yadav. Gamebench: Evaluating strategic reasoning abilities of llm agents.
> arXiv preprint arXiv:2406.06613, 2024
>
> [3] Jonathan Light, Min Cai, Sheng Shen, and Ziniu Hu. Avalonbench: Evaluating llms playing the
> game of avalon. In NeurIPS 2023 Foundation Models for Decision Making Workshop, 2023
>
> [4] Xiao Liu, Hao Yu, Hanchen Zhang, Yifan Xu, Xuanyu Lei, Hanyu Lai, Yu Gu, Hangliang Ding,
> Kaiwen Men, Kejuan Yang, et al. Agentbench: Evaluating llms as agents. arXiv preprint
> arXiv:2308.03688, 2023
>
> [5] Wei Wang, Dan Zhang, Tao Feng, Boyan Wang, and Jie Tang. Battleagentbench: A benchmark for
> evaluating cooperation and competition capabilities of language models in multi-agent systems.
> arXiv preprint arXiv:2408.15971, 2024
>
> [6] Yue Wu, Xuan Tang, Tom M Mitchell, and Yuanzhi Li. Smartplay: A benchmark for llms as
> intelligent agents. arXiv preprint arXiv:2310.01557, 2023
>
> [7] Hendrycks, Dan, et al. "Measuring mathematical problem solving with the math dataset." arXiv preprint arXiv:2103.03874 (2021).
>
> [8] Jimenez, Carlos E., et al. "Swe-bench: Can language models resolve real-world github issues?." arXiv preprint arXiv:2310.06770 (2023).

---

> > ### Author Response · Authors · 2024-11-27
> > **A Decrypto Puzzle**
> >
> > As the manuscript revision period comes to end, we ask whether the reviewer has had a chance to consider our response, our "General Response to all Reviewers", and our updated manuscript. Based on the reviewers' feedback, we have **improved and clarified multiple key points in the paper**, and **included a new experiment measuring two additional ToM abilities** following references provided by reviewer AGZj.
> >
> > If they wish, we also invite the reviewer to try and solve a Decrypto puzzle:
> >
> > ```
> > You play as Eve. Analyse hints from previous rounds and the current hints to intercept the code. For this puzzle, assume all codes of 3 non-repeating digits are possible.
> >
> > Hints from previous rounds:
> > 1. patience, moral, knight
> > 2. money, luck, wheel
> > 3. dog, blue, choke
> > 4. water, oil, pearl
> >
> > Hints from this round:
> > a. neck
> > b. vice
> > c. value
> >
> > What is the code?
> > ```
> >
> > Finally, if we have addressed the reviewer's concerns, we kindly ask them to update their score.

---

> ### Author Response · Authors · 2024-12-02
> **Final day of discussion**
>
> Dear reviewer Z9ie,
>
> Today is the final day of the discussion period. As such, we kindly ask the reviewer if they have any additional feedback following our response. We also ask that they consider increasing their support for our paper if their concerns have been addressed.
>
> Thank you,
>
> The Decrypto Authors

---

### Author Response · Authors · 2024-11-21
**General Response to all Reviewers**

We thank all reviewers for their time and valuable feedback and hope to have addressed all their concerns in our individual responses to them. Their comments have allowed us to greatly improve our work, and as a result we have uploaded a revised version of the paper.

In particular, we have improved writing, clarifying and expanding on key points raised by the reviewers. This includes the following:
- Expanded discussion on the metrics chosen and their justification
- Expanded discussion on the baselines and the design choices we made
- Added details on human data collection, including screenshots of the data collection interface (Appendix B)
- Added examples of LLM failure cases in Appendix C
- Clarified how we create the prompts for the prompt robustness study in Figure 4.

We have also added a **new experiment measuring two additional ToM abilities**, in accordance with the seminal ToM paper by Gopnik & Astington suggested to us by Reviewer AGZj. This experiment measures a strong and weak form of the representational change and false-belief ToM abilities. We show that while ToM abilities correlate with model size, both Llama 3.1 8B and 70B struggle with more refined representational change and modeling of false beliefs, with particularly low scores ($\leq 17\\%$ pass rate) for the strong variant of the tasks.

For the final version of the paper, we also commit to:
- Keep improving the writing
- Collect additional human games from more participants
- Add human interceptor results to Table 1

We hope the changes above encourage reviewers to increase their support for our submission. We also look forward to any additional feedback on how to further improve our work.

Kind regards,\
The Decrypto Benchmark Authors

[1] Gopnik & Astington, Children's Understanding of Representational Change and Its Relation to the Understanding of False Belief and the Appearance-Reality Distinction, 1988

---

### Comment · Area_Chair_CZx9 · 2024-11-25

Dear Reviewers,


This is a friendly reminder that the discussion will end on Nov. 26th (anywhere on Earth). If you have not already, please take a close look at all reviews and author responses, and comment on whether your original rating stands.


Thanks,

AC

---

### Meta-Review · Area_Chair_CZx9 · 2024-12-21

**Metareview:**

The paper introduces Decrypto, a novel interactive benchmark designed to evaluate multi-agent reasoning, coordination, competition, and theory of mind (ToM) capabilities in large language models (LLMs).

- Decrypto introduces a dynamic and interactive framework that goes beyond traditional static input benchmarks.
- The benchmark draws from foundational cognitive science concepts, particularly those used to assess theory of mind.

The weaknesses are as follows.
- Some reviewers argue that Decrypto may not effectively measure theory of mind as defined in cognitive science.
- The human baseline is based on only nine games, which is a very limited sample size, making the comparison less reliable.

**Additional Comments On Reviewer Discussion:**

One reviewer was not engaged in the author rebuttal. Two were engaged, and one of them is still negative. The weaknesses mentioned above are not fully addressed.

---

### Decision · Program_Chairs · 2025-01-22

Reject